# Modeling of waning immunity after SARS-CoV-2 vaccination and influencing factors

Laura Pérez-Alós [1,11✉], Jose Juan Almagro Armenteros[2,11], Johannes Roth Madsen[1], Cecilie Bo Hansen[1], Ida Jarlhelt[1], Sebastian Rask Hamm[3], Line Dam Heftdal [3,4], Mia Marie Pries-Heje[5], Dina Leth Møller [3], Kamille Fogh[6,7], Rasmus Bo Hasselbalch[6,7], Anne Rosbjerg[1], Søren Brunak [2], Erik Sørensen[8], Margit Anita Hørup Larsen[8], Sisse Rye Ostrowski [8,9], Ruth Frikke-Schmidt [9,10], Rafael Bayarri-Olmos [1], Linda Maria Hilsted[10], Kasper Karmark Iversen[6,7,9], Henning Bundgaard[5,9], Susanne Dam Nielsen[3,9] & Peter Garred [1,9✉]

SARS-CoV-2 vaccines are crucial in controlling COVID-19, but knowledge of which factors determine waning immunity is limited. We examined antibody levels and T-cell gamma-interferon release after two doses of BNT162b2 vaccine or a combination of ChAdOx1-nCoV19 and BNT162b2 vaccines for up to 230 days after the first dose. Generalized mixed models with and without natural cubic splines were used to determine immunity over time. Antibody responses were influenced by natural infection, sex, and age. IgA only became significant in naturally infected. A one-year IgG projection suggested an initial two-phase response in those given the second dose delayed (ChAdOx1/BNT162b2) followed by a more rapid decrease of antibody levels. T-cell responses correlated significantly with IgG antibody responses. Our results indicate that IgG levels will drop at different rates depending on prior infection, age, sex, T-cell response, and the interval between vaccine injections. Only natural infection mounted a significant and lasting IgA response.

[1] Laboratory of Molecular Medicine, Department of Clinical Immunology, Section 7631, Rigshospitalet, University of Copenhagen, Copenhagen, Denmark. [2] Novo Nordisk Foundation Center for Protein Research, Faculty of Health and Medical Sciences, University of Copenhagen, Copenhagen, Denmark. [3] Viro-immunology Research Unit, Department of Infectious Diseases, Section 8632, Rigshospitalet, University of Copenhagen, Copenhagen, Denmark. [4] Department of Haematology, Rigshospitalet, University of Copenhagen, Copenhagen, Denmark. [5] The Heart Center, Department of Cardiology, Rigshospitalet, University of Copenhagen, Copenhagen, Denmark. [6] Department of Cardiology, Herlev and Gentofte Hospital, University of Copenhagen, Copenhagen, Denmark. [7] Department of Emergency Medicine, Herlev and Gentofte Hospital, University of Copenhagen, Copenhagen, Denmark. [8] Blood Bank, Department of Clinical Immunology, Section 2034, Rigshospitalet, University of Copenhagen, Copenhagen, Denmark. [9] Department of Clinical Medicine, Faculty of Health and Medical Sciences, University of Copenhagen, Copenhagen, Denmark. [10] Department of Clinical Biochemistry, Rigshospitalet, University of Copenhagen, Copenhagen, Denmark. [11] These authors contributed equally: Laura Pérez-Alós, Jose Juan Almagro Armenteros. ✉email: laura.perez.alos@regionh.dk; peter.garred@regionh.dk

The approved vaccines against SARS-CoV-2 have been shown to be effective and largely safe[1–6]. Randomized clinical trials and real-world data show that the vaccines elicit a rapid and highly protective immune response[1,3], and appeared early on to be effective at different degrees in preventing viral transmission and severe coronavirus disease 2019 (COVID-19)[1–6]. The vaccines elicit a robust response after the second dose among SARS-CoV-2 infection naïve. At the same time, those naturally infected achieve a robust response already after the first vaccine dose[7–10]. However, with the emergence of the B.1.617.2. delta virus variant, it is clear that while the vaccines may be effective in protecting against severe disease, they are less effective in avoiding transmission[11,12]. Concerning the B.1.1.529 omicron variant, the picture seems to be more complicated because it appears to increase infectivity and evade immune recognition by the currently used vaccines[13]. Data suggest that peak viral load in B.1.617.2. delta virus-infected individuals in the airways might not be significantly lowered in vaccinated compared to non-vaccinated individuals with potential importance for onward transmission risk[14]. Breakthrough infections are observed after vaccination, even with detectable antibody levels. However, a protective antibody threshold remains to be established[15]. Moreover, waning immunity six months after vaccination has recently been reported, particularly in older individuals[16].

The importance of antibody and T-cell responses induced by anti-SARS-CoV-2 vaccines is well established[17]. Data suggest that vaccine-induced T-cells respond identically to SARS-CoV-2 variants of concern but differ in longevity and homing properties depending on natural infection[18]. However, real-world data describing the antibody and T-cell kinetic interaction over time are sparse. Thus, a more comprehensive understanding of the difference between natural infection and vaccination and models to foresee waning immunity are urgently needed.

The Danish health authorities halted the use of the ChAdOx1-nCoV19 vaccine from Oxford/AstraZeneca in spring 2021 due to rare side effects (https://www.sst.dk/en/English/Corona-eng). Individuals given the first shot with the ChAdOx1-nCoV19 vaccine were offered a second shot with the BNT162b2 from Pfizer/BioNTech. However, the second injection was in most cases substantially delayed compared to those offered both injections with the BNT162b2 vaccine. The optimal dosing interval for SARS-CoV-2 vaccines remains controversial. However, a delay in the vaccine intervals has been shown to increase peak serum antibody levels[19–21]. Thus, it has been suggested that a delayed second-dose strategy could yield faster partial protection to a larger proportion of the population when vaccine supplies are limited. Nevertheless, whether this will lead to a prolonged sustained response is unknown.

The mucosal immune system, with its main component IgA, is the largest component of the entire immune system[22]. IgA and mucosal immunity have evolved to provide first-line protection at the main entry of infectious threats[22]. As SARS-CoV-2 initially infects the upper respiratory tract, its first interactions with the immune system occur predominantly at the respiratory mucosal surfaces where IgA is the predominant immunoglobulin, which may be linked to sterilizing immunity[23]. The current vaccines are administered intramuscularly and elicit high IgG and neutralizing antibody responses 7–14 days after receiving the second vaccine injections[24]. IgA responses are as well observed and although levels can last weeks they may wane faster than IgG after vaccination[25,26]. Circulating IgA antibodies might provide additional information of vaccine responses[27], but whether this can be translated into natural mucosal protection in the upper airways and the lungs and the development of sterilizing immunity is not clear. Nevertheless, it is conceivable that a robust IgA response is necessary to obtain early sterilizing immunity, as observed in natural infection primed individuals[23].

In this study, we constructed models to study the vaccine response kinetics using consecutive serological and cellular data from a large population of vaccinated health care professionals. We could model the IgG response and provide an estimation of IgG levels up to one year after vaccination. Additionally, we describe antibody neutralization and IgM and IgA serum responses using these data. This allowed us to investigate the importance of natural infection, vaccine timing, and accompanying T-cell responses on the level of the main immunoglobulin classes and antibody neutralization capacity.

## Results

**Characteristics of the study population.** A total of 1754 individuals participated in the study (1516 (86.4%) were females) at a median age of the entire cohort of 49 (IQR: 39–58) years (Table 1). Blood samples were collected prior to vaccination and up to 230 days after the first dose and subjected to analysis for SARS-COV-2 nucleocapsid (N) protein and receptor-binding domain (RBD) seropositivity and neutralizing antibodies determination. The number of repeated measurements per participant included in the study varied from 3 to 5. By determining the seropositivity for N protein antibodies, we identified 161 individuals (140 females, 87%) who were primed with SARS-CoV-2 before the administration of the first or the second dose (Table 1). Among all the participants who were N protein-positive (n = 161), and thus naturally infected with SARS-CoV-2, 91 (56.5%) individuals had a positive result by the reverse transcription-polymerase chain reaction (RT-PCR) before the first dose, while 14 (8.7%) received it between the first and second vaccine dose (Supplementary Table 1). The majority of participants were offered the BNT162b2 vaccine (n = 1,634, 93.2%), while a group of 120 individuals received the ChAdOx1/ BNT162b2 vaccine (Table 1). Participants who were vaccinated

**Table 1 Demographic data and characteristics from the study cohort.**

| | N protein negative (N = 1593) | N protein positive (N = 161) | Total (N = 1754) |
|---|---|---|---|
| Sex | | | |
| Female | 1376 (86.4%) | 140 (87.0%) | 1516 (86.4%) |
| Male | 217 (13.6%) | 21 (13.0%) | 238 (13.6%) |
| Age (years) | | | |
| Median (IQR) | 50 (39–59) | 44 (31–54) | 49 (39–58) |
| <40 | 441 (27.7%) | 70 (43.5%) | 511 (29.1%) |
| >40–60 | 850 (53.4%) | 78 (48.4%) | 928 (52.9%) |
| >60 | 302 (19.0%) | 13 (8.1%) | 315 (18.0%) |
| BMI[a] | | | |
| Median (IQR) | 24 (22–27) | 24 (22–27) | 24 (22–27) |
| Underweight (<18.5) | 20 (1.3%) | 3 (1.9%) | 23 (1.3%) |
| Normal (18.5–24.9) | 806 (50.6%) | 87 (54.0%) | 893 (50.9%) |
| Overweight (25–29.9) | 374 (23.5%) | 42 (26.1%) | 416 (23.7%) |
| Obese (>30) | 206 (12.9%) | 18 (11.2%) | 224 (12.8%) |
| Vaccine administrated | | | |
| BNT162b2 | 1506 (94.5%) | 128 (79.5%) | 1634 (93.2%) |
| ChAdOx1 nCoV19/ BNT162b2 | 87 (5.5%) | 33 (20.5%) | 120 (6.8%) |
| Days between doses (days) | | | |
| BNT162b2 | 30 (29–32) | 31 (29–33) | 30 (29–33)[b] |
| ChAdOx1 nCoV19/ BNT162b2 | 81 (80–83) | 82 (81–83) | 81 (80–83)[b] |

[a]Missing data (n = 198).
[b]Mann–Whitney U test (two-sided), p < 2.2e−16).

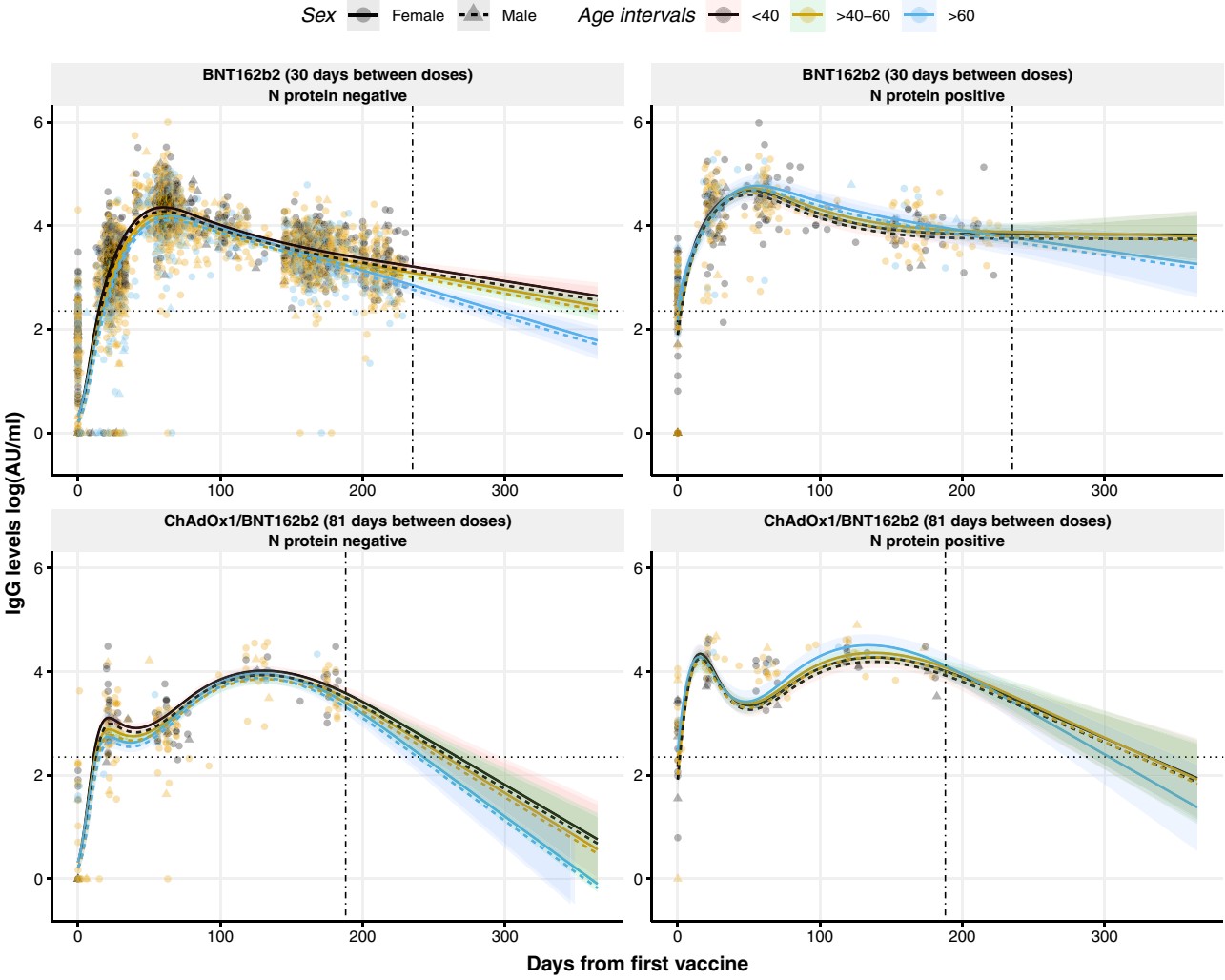

**Fig. 1 Dynamics and projection of circulating IgG levels against RBD after the first dose of the COVID-19 vaccination using a non-linear model.**
Distribution of IgG levels, represented in log(AU/ml), over time (days from the first vaccine) in individuals with no prior infection vaccinated with BNT162b2 (top left) or with the combination ChAdOx1/BNT162b2 (bottom left), and in individuals previously infected and vaccinated with BNT162b2 (top right) or with the combination ChAdOx1/BNT162b2 (bottom right). Circles and triangles represent the observed levels of circulating IgG antibodies in females and males, respectively. Solid and dashed lines represent the predicted levels of circulating IgG antibodies calculated by the model in females and males, respectively. Black, yellow, and blue colors represent individuals with age <40, 40–60, and >60 years, respectively. Horizontal black dotted line represents the threshold for assay positivity. Vertical dash-dotted line indicates where the out-of-sample trend starts. Shadowed areas represent the 95% confidence interval. Center for the confidence interval is the predicted (mean) values.

with two doses of BNT162b2 received the second dose after a median of 30 (IQR: 20–33) days, a shorter period than those who received the ChAdOx1/BNT162b2 vaccine combination, which was 81 (80–83) days (Mann–Whitney U test, $p < 0.0001$) (Table 1). Supplementary Figure 1 illustrates a timeline specifying sample collection and dose administration.

**Dynamics of circulating IgG levels over time after vaccination.**
Levels of IgG against RBD upon BNT162b2 vaccination fluctuated over time ($p < 0.0001$) (Fig. 1, top panels), where the IgG levels increased exponentially until reaching a maximum approximately at day 60 after the first dose and hereafter decreased over time. For individuals who received the ChAdOx1/BNT162b2 vaccine combination, the IgG kinetics were different from the ones observed for the BNT162b2 vaccine ($p < 0.0001$) (Fig. 1, bottom panels). The IgG levels reached the first peak around day 20 with a second peak observed around day 130. The IgG levels of the latter peak were similar to the peak value

achieved in the BNT162b2 vaccinated group. However, this increase was followed by a rapid decline compared to BNT162b2 vaccinated individuals. This decline was more pronounced in those previously non-infected.

An interaction between days from the first vaccine dose and age was found, with the older age group (>60 years old) presenting a more marked decrease of IgG levels compared to the younger groups ($p < 0.0001$). Consistent with our previous results[7], infection with SARS-CoV-2 prior to the complete vaccination was associated with higher IgG levels over time compared to non-naturally infected individuals ($p < 0.0001$) (Fig. 1). Furthermore, the male sex was associated with lower IgG levels over time ($p = 0.0012$). An interaction between N protein seropositivity and age was observed ($p = 0.0044$), where naturally infected individuals above 40 years old had higher IgG antibody levels at the second dose compared to the younger individuals. On the other hand, no associations were observed between IgG levels and body mass index (BMI) (Supplementary Fig. 2).

By using a generalized mixed model with four natural cubic splines (Fig. 1), we developed a non-linear model to quantitatively determine the levels of circulating IgG levels at any time point after vaccination taking into consideration the sex and the age groups of each individual (<40, 40–60, >60 years old), the evidence of prior infection (N protein seropositivity), and the vaccine administrated. To provide an estimation on how the IgG levels would develop beyond the range of the data (230 days), the estimated trends were projected up to one year after vaccination. This allowed us to provide personalized IgG level dynamics useful to find eligible candidates for re-vaccination. For example, for a female individual, between 40 and 60 years old, with previous SARS-CoV-2 infection, the estimated IgG levels after 60 days from vaccination with BNT162b2 is 49,888 AU/ml (95% confidence interval (CI): 40,330–61,308 AU/ml), projected to decrease in 365 days after the first dose to 6279 AU/ml (95% CI: 2133–18,950 AU/ml). When comparing these predicted values with a female from the same age and vaccine group without prior infection, the estimated levels at 60 and 365 days after vaccination would be 17,408 AU/ml (95% CI: 16,204–18,734 AU/ml) and 282 AU/ml (184–431 AU/ml), respectively, reflecting the great influence of previous SARS-CoV-2 infection in the development of a more stable IgG response over time after vaccination. This means, within a timeframe of 305 days, with natural SARS-CoV-2 infectious priming, the IgG levels would decrease in general with around 87% from the peak value for the BNT162b2 alone, while for the ChAdOx1/BNT162b2 combination the IgG levels decrease close to 97% from the peak value. Individuals without prior natural SARS-CoV-2 infectious priming would lose around 98 and 100%, respectively, according to our model. However, the decline differs to some degree according to age and sex (Fig. 1). In addition, we represented the prediction intervals (PI) for the same model to illustrate the uncertainty of future values (Supplementary Fig. 3).

We also constructed a linear model that quantitively determines IgG levels from the second vaccine dose (Fig. 2), which considers the same factors as the generalized mixed model with four natural cubic splines. Contrary to the non-linear model, the IgG kinetics from the second dose does not differ significantly between vaccines. The same is observed for the different age groups, which do not influence significantly. Following the above-mentioned example to provide personalized IgG levels, it is observed a significant influence of the previous infection in the stability of the IgG response over time ($p < 0.0001$) albeit the decline is more pronounced. Within the same timeframe of 305 days (from day 20 after the second dose projected to day 325, which is a similar range compared to the non-linear model), with natural infection priming IgG levels would decrease close to 99 and 97% for the BNT162b2 alone and the ChAdOx1/BNT162b2 combination, respectively. Infection naïve individuals would lose around 99% IgG levels for both vaccine combinations administered.

**IgM and IgA responses after vaccination**. While IgG observations only had a high proportion of zeros at baseline, IgM and IgA observations had a high proportion of zeros across all the time points (Supplementary Fig. 4). This broke the assumptions of a normally distributed response variable thus producing very skewed non-normally distributed residuals. Therefore, we decided to model the IgA and IgM levels as binary variables. Binomial generalized mixed models (positive/negative outcome) were employed to study the IgM responses against RBD after vaccination (Supplementary Fig. 5). As observed for IgG, a natural infection prior to vaccination increased the probability of an IgM positive response. The positivity of IgM should be carefully

interpreted as the IgM levels are rather low, close to the positivity threshold.

To study IgA responses, binomial generalized mixed models were used as well. Figure 3 depicts the distribution of IgA positive responses from baseline until the end of the study. It is evident that individuals previously infected displayed an increased percentage of IgA positive responses as the probability of positive responses was greater and maintained over time compared to those with no evidence of natural infection ($p = 0.0413$). For example, the estimated probability of positive response after 180 days from vaccination in a female between 40 and 60 years old primed with the virus would be, according to our model, 75% (95% CI: 60–86%) compared to 4% (95% CI: 3–6%) without previous infection. These kinetics are in line with previous reports measuring IgA against S1 protein[25,26]. Interestingly, in non-infectious primed individuals fully vaccinated with BNT162b2, there was a peak of positive responses around day 60 after the first dose, whereas this was not observed in non-primed individuals vaccinated with ChAdOx1/BNT162b2 ($p < 0.0001$). As an example, a female between 40 and 60 years old vaccinated solely with BNT162b2, at day 60, the estimated probability of a positive IgA response would be 57% (95% CI: 51–63%), whereas for a female of the same age but administrated the ChAdOx1/BNT162b2 combination, the estimated probability at day 60 would be 5% (95% CI: 2–11%).

**Influence of time intervals between the first and the second dose**. Individuals vaccinated with ChAdOx1/BNT162b2 developed slightly lower levels of IgG antibodies after administration of the second dose and showed a faster decline compared to those vaccinated with BNT162b2 two times (Fig. 1, $p < 0.0001$). As an example, the projected levels of IgG antibodies for a non-primed female between 40 and 60 years old vaccinated with BNT162b2 after 250 days from the first dose would be 1038 AU/ml (95% CI: 871–1239 AU/ml), whereas when vaccinated with ChAdOx1/BNT162b2 the projected IgG levels at the same timepoint would be 312 AU/ml (148–653 AU/ml). Based on the different IgG dynamics between both vaccines, which have a different dosing interval, we investigated the effect at the moment of administration of the second dose in individuals fully vaccinated with the BNT162b2 vaccine to evaluate if the time interval between the two doses was associated with altered antibody levels. Supplementary Figure 6 displays development differences of IgG levels in individuals with an interval of <29 days between doses compared to those with >29–31 and >31 days between doses ($p < 0.0001$). Here, we observe a decrease in IgG levels at the peak value (day 60) of 14% in individuals with a vaccine interval >31 days compared to those with a vaccine interval <29 days. Moreover, after 230 days from the first dose, the antibody difference between these groups is close to 9%. These observations were more evident in naturally infected individuals, where the difference at the peak was 50% and after 230 days was close to 47% (Supplementary Fig. 6). A similar pattern was observed for IgA responses ($p = 0.0130$), being more evident in previously infected individuals, supporting the role of IgA in the development of a robust response (Supplementary Fig. 7). This suggests that longer intervals between vaccines appear to mount a less robust antibody response causing a more rapid decline in IgG and IgA concentrations.

**Neutralizing antibodies (nAbs) against SARS-CoV-2**. We used a binary model to evaluate the capacity of the circulating antibodies to neutralize the virus (Fig. 4). As observed for IgG, male sex was associated with a significantly lower response of nAbs compared to females ($p = 0.0084$). As seen for IgG, previous

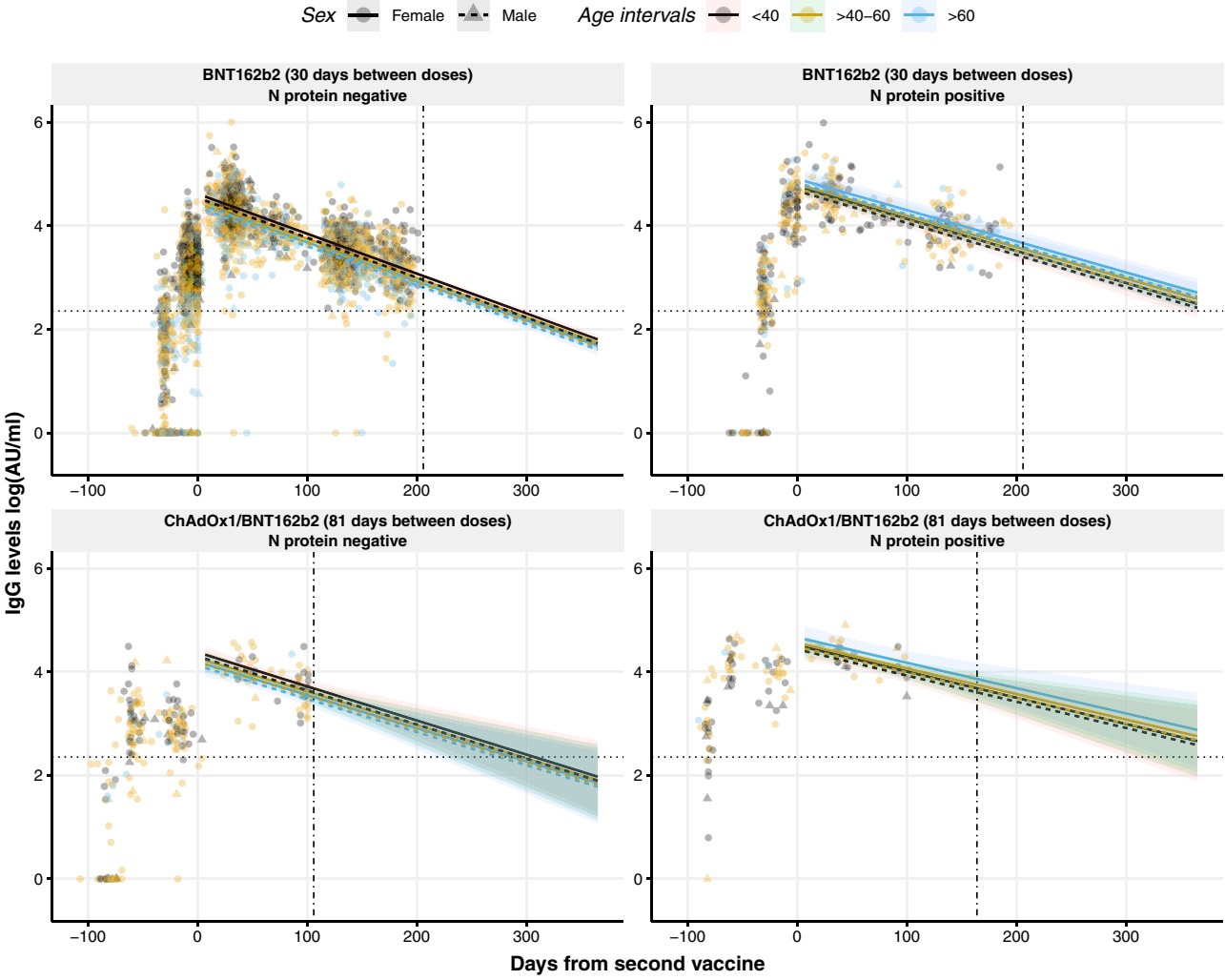

**Fig. 2 Dynamics and projection of circulating IgG levels against RBD after the second dose of the COVID-19 vaccination using a linear model.**
Distribution of IgG levels, represented in log(AU/ml), over time (days from the second vaccine) in individuals with no prior infection vaccinated with BNT162b2 (top left) or with the combination ChAdOx1/BNT162b2 (bottom left), and in individuals previously infected and vaccinated with BNT162b2 (top right) or with the combination ChAdOx1/BNT162b2 (bottom right). Circles and triangles represent the observed levels of circulating IgG antibodies in females and males, respectively. Solid and dashed lines represent the predicted levels of circulating IgG antibodies calculated by the model in females and males, respectively. Black, yellow, and blue colors represent individuals with age <40, 40–60, and >60 years, respectively. Horizontal black dotted line represents the threshold for assay positivity. Vertical dash-dotted line indicates where the out-of-sample trend starts. Shadowed areas represent the 95% confidence interval. Center for the confidence interval is the predicted (mean) values.

infection was associated with the development of faster nAbs responses ($p < 0.0001$). Moreover, an interaction between age and previous infection was observed, where younger individuals (<40 years old) developed faster responses compared to the participants above the age of 40 years ($p < 0.0001$). Finally, the stability of the nAbs responses at the end of the study was notable as practically all individuals had positive nAbs responses regardless of the vaccine and previous natural infection. Since the assay becomes fully saturated, the dynamic range of the assay might be limited.

**Correlation between cellular response and humoral response.**
We analyzed 250 samples collected between 90 and 230 days after the first dose to evaluate the cellular response by measuring the levels of IFN-γ from activated T-cells upon recognition of S1 peptides. As shown in Supplementary Fig. 8C, increasing levels of IFN-γ correlated positively with IgG levels after 6 months (Spearman rank, $\rho = 0.43$, $p < 0.0001$), but also in the same range

with the IgG titer at sampling points after 21 days and 2 months ($\rho = 0.38$ and $\rho = 0.37$, respectively) (Supplementary Fig. 8A, B, respectively). On the contrary, an increase in IgA responses was not associated with IFN-γ levels (Supplementary Table 2). As described for IgG antibodies, age was negatively associated with IFN-γ levels ($p = 0.0365$ and $p = 0.0332$), whereas the previous infection was associated with higher IFN-γ levels released from activated T-cells ($p = 0.0028$) (Supplementary Table 2). Interestingly, when the IFN-γ levels were divided into low, intermediate, or high levels and used to model the IgG levels over time (Fig. 5) it was observed that individuals with higher IgG response and slower antibody decrease over time had higher IFN-γ levels ($p = 0.0046$).

## Discussion
Most of the studies on COVID-19 vaccine responses have been performed on vaccine effectiveness, which is based on test-negative, case-control study design and does not reflect the humoral

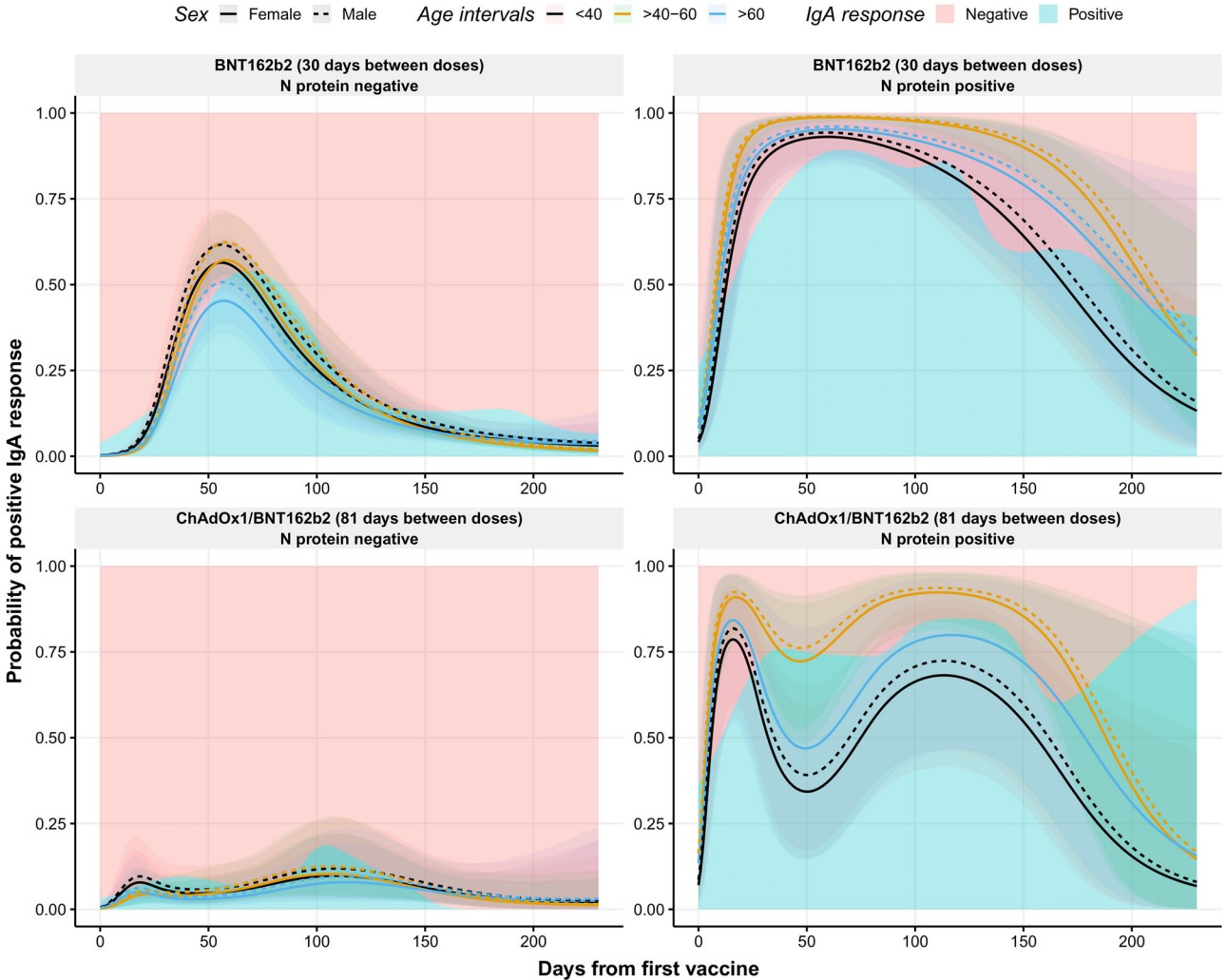

**Fig. 3 Observed and predicted probability of positive IgA responses against RBD after the first dose of the COVID-19 vaccination.** Distribution of positive IgA response (probability) over time (days from the first vaccine) in individuals with no prior infection vaccinated with BNT162b2 (top left) or with the combination ChAdOx1/BNT162b2 (bottom left), and in individuals previously infected and vaccinated with BNT162b2 (top right) or with the combination ChAdOx1/BNT162b2 (bottom right). Blue and pink backgrounds represent the conditional density estimation of positive and negative IgA responses, respectively. Solid and dashed lines represent the predicted probability of positive IgA responses calculated by the model in females and males, respectively. Black, yellow, and blue colors represent individuals with age <40, 40–60, and >60 years, respectively. Shadowed areas represent the 95% confidence interval. Center for the confidence interval is the predicted (mean) values.

response in terms of antibody levels[28]. On the other hand, few publications based on follow-up samples measuring humoral responses after COVID-19 vaccination on real-world data have reported IgG levels and/or neutralizing antibodies against SARS-CoV-2, and the problem of waning immunity[16]. Here, we report a complete overview of the humoral responses based on IgM, IgG, and IgA analyses and the neutralizing antibodies from health care professionals since the first vaccine dose and up to 230 days after. From a smaller group of the participants, we show the T-cell response to vaccination at approximately 6 months after vaccination. We have developed a non-linear model using a generalized mixed model with natural cubic splines, which considers the age, previous infection, sex, the vaccine administrated, and the day after the first SARS-CoV-2 vaccine dose. This model represents the current situation observed on antibody waning[16], and projects the levels of IgG antibodies up to one year from first vaccination, revealing that the IgG antibody levels would decrease by around 87–100%, depending on the prior natural infection, age, sex and time between first and second vaccination. Nonetheless, we should be cautious about the projections beyond the range of the data.

Natural cubic splines are not optimal for the extrapolation task. However, they are able to provide a linear extrapolation beyond the range of the data, which facilitates the understanding of how the IgG levels could develop in the future. We do not discard the possibility of the antibody waning taking different shapes than the one outlined by our model. Potentially, this waning could stabilize over time or experience a sharper or slower decrease. Also, many of these individuals will encounter SARS-CoV-2 variants or get a booster shot in this period of time, which will undoubtedly modify the dynamics of antibody waning. Our model differs from other recently published models as we include the time from the first vaccine dose into the model fit[16]. This allows us to evaluate the baseline levels before the administration of the vaccine, as at this time point, the difference in IgG levels between previously infected and not-infected individuals has a greater influence than after the administration of the second dose. As the dynamics of IgG levels over time are not linear, our model is better designed for this purpose.

Linear mixed models were considered, but due to the clear non-linear trend of the antibody levels, we decided to model time

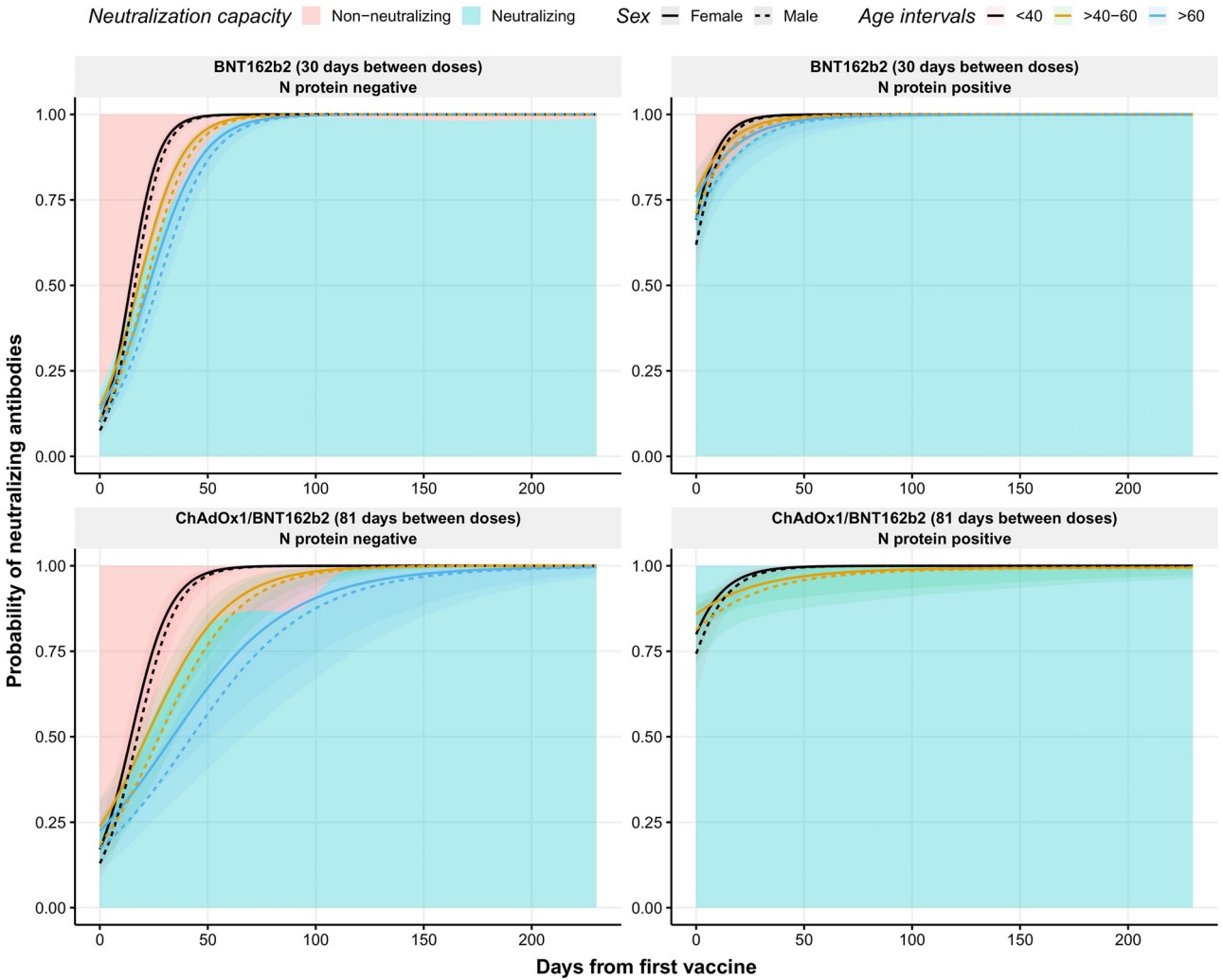

**Fig. 4 Observed and predicted the probability of neutralizing antibodies against RBD after the first dose of the COVID-19 vaccination.** Distribution of neutralizing antibodies (probability) over time (days from the first vaccine) in individuals with no prior infection vaccinated with BNT162b2 (top left) or with the combination ChAdOx1/BNT162b2 (bottom left), and in individuals previously infected and vaccinated with BNT162b2 (top right) or with the combination ChAdOx1/BNT162b2 (bottom right). Blue and pink backgrounds represent the conditional density estimation of neutralizing and non-neutralizing antibodies index, respectively. Solid and dashed lines across the graph represent the predicted probability of neutralizing antibodies index calculated by the model in females and males, respectively. Black, yellow, and blue colors represent individuals with age <40, 40–60, and >60 years, respectively. Shadowed areas represent the 95% confidence interval. Center for the confidence interval is the predicted (mean) values.

using natural cubic splines. However, for comparison with previously published studies[16] and for clarity reasons, we also constructed linear models that quantitively determine IgG levels from the second vaccine dose and took into account the same factors as the generalized mixed model with four natural cubic splines. In contrast to the non-linear model, the IgG kinetics from the second dose did not differ significantly between vaccines. Infection naïve individuals exhibit a linear decay of IgG levels, and both linear and non-linear models presented similar estimations. On the other hand, the linear and non-linear models disagreed with IgG levels dynamics of previously infected individuals. This might indicate that IgG kinetics for these individuals are more complex and a non-linear model is better suited for this task. When fitting the mixed model with natural cubic splines for the IgG levels, we realized that the assumption of the model (Gaussian distribution of the response variable) was not met. Model diagnostics showed non-normally distributed residuals and a high degree of heteroscedasticity. This was due to the high density of zero log10-transformed IgG levels (below the detectable limit) for the baseline observations. We utilized a zero-inflated Gaussian mixed

model to account for these observations while modeling the non-zero observations, which were normally distributed. This two-part model allowed to jointly fit the two distributions present in the data, which was reflected in a better fit and correct model diagnostic. The IgM and IgA levels and neutralization index were not normally distributed along with the time variable. This differed from IgG, where the response was non-normally distributed only at baseline. These differences arise due to the overall lower levels of IgM and IgA, which makes them more difficult to detect not only at baseline. On the other hand, the neutralization index rapidly increased from 0 to 100 and levels remained close to 100 overtime, leading to a distribution with most of its mass at 0 and 100. The common factor of IgM and IgA levels and neutralization index was that they were following a binomial distribution. IgM and IgA levels were then modeled as positive or negative response and neutralization index as neutralizing power or not.

The results of this real-world longitudinal study show the different dynamics of antibody levels over time in individuals vaccinated with either BNT162b2 or ChAdOx1/BNT162b2 vaccines. Following other reports, individuals who were infected

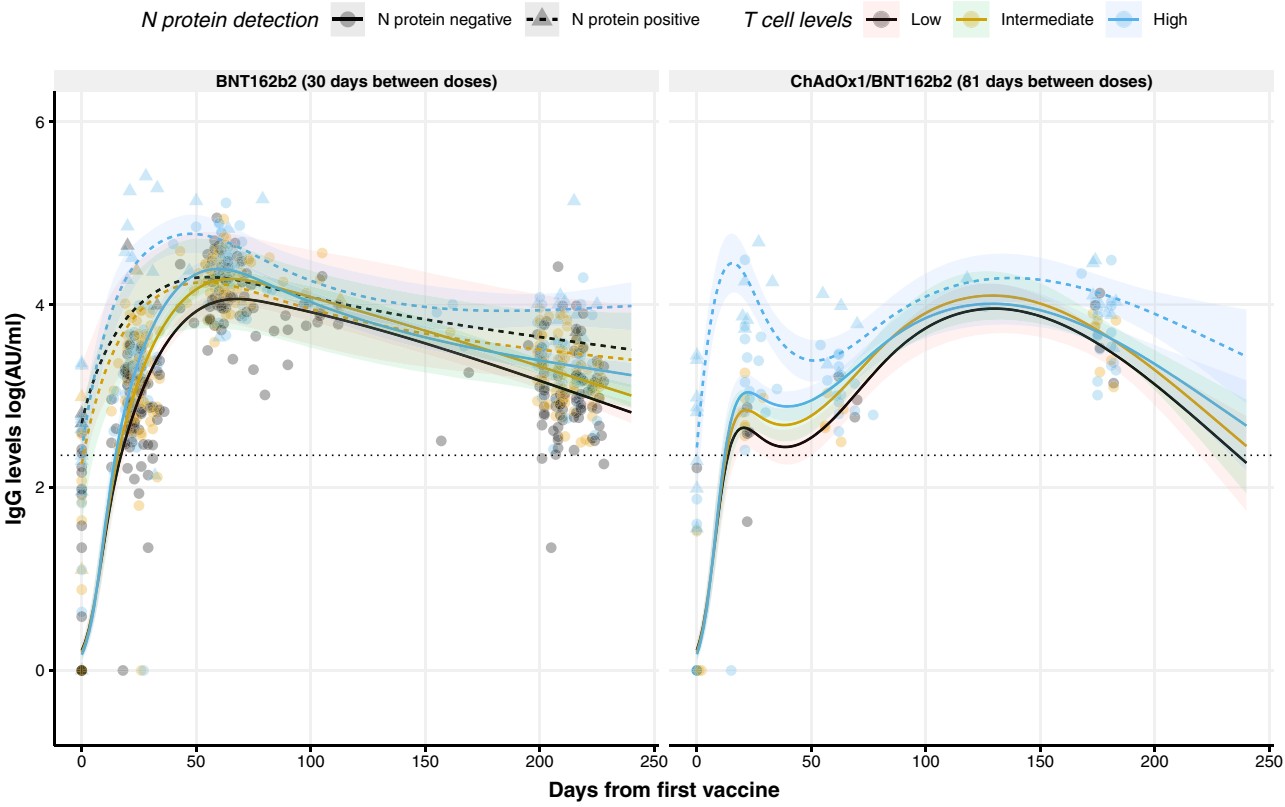

**Fig. 5 Circulating IFN-γ levels modeled to circulating IgG levels against RBD in individuals vaccinated against SARS-CoV-2.** Distribution of circulating IgG levels, represented in log(AU/ml), over time (days from the first vaccine) in individuals vaccinated with BNT162b2 (left) or with the combination ChAdOx1/BNT162b2 (right). Circles and triangles represent the observed levels of circulating IgG levels for non-previously infected and infected individuals with SARS-CoV-2, respectively. Solid and dashed lines represent the predicted levels of circulating IgG levels for non-previously infected and infected individuals with SARS-CoV-2, respectively. Black, yellow, and blue colors represent low, intermediate, and high levels of IFN-γ released from stimulated T-cells against peptides derived from the S1 subunit of S protein. Black dotted line represents the threshold for assay positivity. Shadowed areas represent the 95% confidence interval. Center for the confidence interval is the predicted (mean) values.

prior to vaccine scheme completion generated higher levels of IgG antibodies. These appeared to remain stable over time, decreasing at a slower rate compared to non-previously infected individuals[10]. Interestingly, age was significantly correlated to a higher level of IgG antibodies in previously infected individuals. This can be explained by the increase in circulating antibodies in older individuals after infection as they are more susceptible to a more severe course of the disease, as we previously reported[29]. Despite this observation, older individuals (>60 years old), displayed a notable decrease in IgG levels independently of the previous infection. This suggests the urgent need to administrate a third dose to the elderly population in the current situation threatened by new breakthrough infections among vaccinated individuals[15]. Regardless of the decrease in IgG levels, the neutralizing capacity of circulating antibodies remained high, indicating the high efficiency of antibodies induced by vaccination. Although the pseudo-neutralization is limited to assay dynamics, we interpret this discrepancy with affinity maturation and increased avidity of the vaccine antibodies occurring over time as has been shown for COVID-19 antibodies and viral neutralization[30,31].

We observed differences in the IgG kinetics between the BNT162b2 and ChAdOx1/BNT162b2 vaccines, where the latter generated a delayed peak of IgG levels, corresponding with the prolonged period before the administration of the second dose for those who received ChAdOx1 as an initial option. Even though both vaccines reached similar peaks of IgG levels, which is

consistent with studies showing even higher antibody levels after a delay of giving the second dose[19,20], our non-linear model estimated an accelerated decline of these circulating antibodies after a year from vaccination, independently of age, sex, or previous natural infection. When comparing vaccine intervals in individuals fully vaccinated with BNT162b2, we observed a significant difference in the IgG peak levels and waning associated with longer intervals between vaccines. This might indicate an influence of the postponement in the vaccine dosing interval, being more evident for infection primed individuals. Nevertheless, this statement should be interpreted carefully as there might be a confounding effect between the vaccine types and intervals between doses. The differences that we observe between individuals administered ChAdOx1 as a first dose and individuals administered BNT162b2 as a first dose cannot be safely attributed to either the vaccine type or the length of the interval between doses. However, for natural infection primed individuals fully vaccinated with BNT162b2 appear to be an indication that a delay in the administration of the second dose has an influence in the peak generated and a more stable long-term humoral response, which should be considered when planning vaccine campaigns.

The most abundant antibody in the mucosal sites is IgA and it has been associated with virus neutralization in SARS-CoV-2 patients[32]. Although the serum and mucosal IgA do not have the same source and; therefore, have different roles, the measurement of circulating IgA antibodies provides further information on vaccine responses[27]. We observed that serum IgA responses were

boosted in previously naturally infected individuals who received any of the two COVID-19 vaccines included in our study. This is an expected feature as the virus main entry is the upper respiratory mucosal tract, inducing mucosal-derived IgA. The circulating IgA from non-primed individuals could originate from bone marrow. However, other production sites are possible[33]. We cannot confidently say which body compartment originates the elevated and lasting serum IgA from in infectious primed individuals. Still, the notable difference in IgA responses between the two groups likely reflects differences between mucosal viral exposure in the airways and systemic vaccine exposure. Nevertheless, an increased peak was noticed in non-previously infected individuals after 60 days from BNT162b2 vaccination. This could be a sign of an improved immune response to the vaccine when no previous infection has occurred that might be related to the shorter interval between doses, as the IgA peak appeared sooner in those individuals who were administered earlier with the second dose.

Cellular responses were measured in 250 of the included participants evaluated between 90 and 230 days after the first dose, and we found that 81% had a positive T-cell reactivity against SARS-CoV-2 (Supplementary Fig. 9). T-cell activation upon viral antigen recognition is crucial for the maturation of the B-cells to mount efficient antibody responses, and thus, a correlation between IgG and T-cell responses has been reported[34]. We showed that the levels of IFN-γ released correlated with IgG levels, including at the initial stages after the administration of the vaccine despite the delayed time of sampling. This indicates that initial T-cell responses might be helpful to predict the evolution of the immune response.

Although the majority of the population in many developed countries are fully vaccinated, increasing infection rates among vaccinated individuals is currently a matter of concern[16]. The dominance of the B.1.617.2 variant is especially a major concern, which has shown similar viral loads in the upper respiratory tract in vaccinated and not vaccinated individuals[14]. One of the factors behind these infections might be the lack of IgA responses after vaccination without prior infection as the approved vaccines are administered intramuscularly. However, IgA response appears to be less robust than natural mucosal protection when vaccines are administrated by this route. This is in accordance with results observed in the present study since infection naïve individuals vaccinated with BNT162b2, mounted an initial IgA response which rapidly disappeared. Those vaccinated with the ChAdOx1/BNT162b2 combination hardly achieved any IgA response, again emphasizing the importance of timing. On the other hand, naturally infected individuals showed a more stable IgA response overtime prior to the BNT162b2 vaccine administration, suggesting a role of IgA to obtain sterilizing immunity.

It is pertinent to address some limitations of the study. In the health care professional population, there is a predominance of females that might skew the study. We did not have access to possible comorbidities and data on alcohol and smoking habits that could influence the immune status and information about the detailed occupation in the hospital of the participants nor the symptomatology of infectious-primed individuals. The prediction models are only valid for apparently healthy individuals and not patients with concomitant diseases that may suppress the immune system. Studies are also needed to investigate if the models are valid using whole S protein or S1 or S2 domains as antigen. A future possibility would be to compare the response against SARS-CoV-2 variants of concern RBDs as delta and omicron. Because of the vaccines used in the present cohort, we could not test other approved vaccines from Moderna and Johnson & Johnson. The study did not have power enough to evaluate the interaction between immunological parameters,

transmission, and disease during the study period. We only studied T-cell activity at approximately 6 months after vaccination, which thus will require follow-up studies to provide firm conclusions. However, the strengths of this study include the comprehensive large-scale use of a broad spectrum of analyses covering different aspects of SARS-CoV-2 immunity and long-term follow-up data allowing the construction of robust models of waning immunity.

There is no consensus on the establishment of a standardized threshold for immune protection against SARS-CoV-2; however, it has been suggested that a combination of different factors combining humoral and cellular responses might help to provide the information to base the definition of protection[35]. Based on our results, there might be an indication that an individual is protected when (a) a certain level of IgG and IgA responses are detected, (b) a correlation between IgG and neutralizing antibodies is present, (c) and activated T-cells upon antigen recognition is observed. In addition to observational results, modeling antibody kinetics appears to be a crucial technology to evaluate which population might suffer from waning immunity and thus need an immunological boost. Importantly, this study does not account for the strength of obtained memory B-cell responses, thus we cannot exclude that this is a crucial factor for the observed breakthrough infections among vaccinated[36–38]. Thus, easy assays suitable for large-scale monitoring of memory B-cell responses than those available at present are needed.

In conclusion, humoral responses following COVID-19 vaccination decreased in all age groups after approximately 6 months. Non-linear models estimated a continuous waning in humoral responses primarily in the older population and individuals with a delayed administration of the second vaccine dose. Natural infections prior to completion of vaccination induced a more robust immune response, characterized primarily by the stability of the IgG levels, higher levels of IFN-γ release from activated T-cells, and the presence of IgA antibodies in circulation. The use of dynamic models taking into consideration different factors of the immune system may be of use to establish guidelines for future re-vaccination strategies.

## Methods

**Study design and participants**. Health care professionals from Rigshospitalet and Herlev-Gentofte University Hospital (Capital Region of Denmark) volunteered to participate in this approximately 6-months prospective longitudinal observational study to determine the dynamics of the antibody levels after SARS-CoV-2 infection and/or COVID-19 vaccination. The sample collection strategy did not interfere with the Danish COVID-19 vaccination program. Participants received either two doses of the BNT162b2 COVID-19 vaccine from Pfizer-BioNTech (BNT162b2) or the combination of one dose of the ChAdOx1-nCoV19 from Oxford/AstraZeneca followed by one dose of the BNT162b2 COVID-19 vaccines (ChAdOx1/BNT162b2). Samples were collected before or up to 14 days after the first dose and approximately 3 weeks, 2 months, and 6 months after the first dose. Age, sex, weight, and height were collected from online questionnaires completed by the participants. Venous blood samples collection fulfilled the principles described in the Declaration of Helsinki. The participants provided informed consent after oral and written information. The protocol was approved by the Regional Scientific Ethics Committee of the Capital Region of Denmark (H-20079890).

**Determination of antibody levels**. Levels of IgM, IgG, and IgA against SARS-CoV-2 spike (S) protein receptor-binding domain (RBD) were measured using a nationally validated in-house ELISA-based assay as described elsewhere with minor modifications[29]. Briefly, 1 μg/ml of recombinant (r) RBD produced in-house was coated onto Nunc-Maxisorp 384-well plates (464718; Thermo Fisher Scientific) in PBS overnight (ON) at 4 °C. Plates were blocked with PBS and 0.05% Tween (Merck) (PBS-T) for 1 h. Subsequently, diluted serum samples were applied in dilution buffer and incubated for 1 h. Antibodies were detected by adding 0.5 μg/ml HRP-conjugated polyclonal rabbit anti-IgM, IgG, or IgA (P0215, P0214, and P0216, respectively, all from Agilent Technologies) and incubated for 1 h. Plates were developed for 7 min (IgG) or 10 min (IgM, IgA) using TMB-One (4380A, Kem-En-Tec) as a substrate. The reaction was stopped using 0.3 M $H_2SO_4$. A mixture containing equal concentrations of r-human IgM, IgG, and IgA antibodies against S protein/RBD was used as a calibrator (1:2000 dilution; A02046-100,

A02038-100, and A02071-100, respectively, all from Genscript). The optical density (OD) was measured using a Synergy HT absorbance reader (Biotek Instruments) at 450–630 nm. Between steps, the plates were washed with PBS-T four times. All incubation steps were performed at room temperature (RT) shaking. The threshold for assay positivity was set to 200, 225, and 100 AU/ml for IgM, IgG, and IgA, respectively.

Elecsys® Anti-SARS-CoV-2 assay (Roche Diagnostics) was employed as a proxy for determining previous natural infection with SARS-CoV-2 by detecting total antibodies against the nucleocapsid (N) protein. This electrochemiluminescence immunoassay (ECLIA) was performed on the COBAS 8000 platform (e801 module) following the manufacturer's instructions.

**Virus-neutralizing antibodies measurement**. An in-house ELISA-based pseudo-neutralizing assay that measures the interaction between the ACE-2 host receptor and RBD was used to estimate the degree of inhibition of virus-neutralizing antibodies against RBD as described previously[39]. Shortly, Nunc Maxisorp 96-well plates (442404; Thermo Fisher Scientific) were coated with 1 µg/ml in-house-produced r-ACE-2 ectodomain ON at 4 °C. Diluted serum samples at 1:10 were incubated with a mixture containing HRP-conjugated high-sensitivity streptavidin (1:16,000 dilution; 21130, Pierce) and 4 ng/ml biotinylated r-RBD (all diluted in PBS-T) in Nunc low-binding round-bottom 96-well plates (Thermo Fisher Scientific) for 1 h at RT. Subsequently, this solution was transferred to ACE-2-coated plates and incubated for 35 min at RT. Plates were developed for 20 min as described above. Between steps, the plates were washed with PBS-T three times. All incubation steps were performed at RT shaking. The threshold for assay positivity was set to 25% inhibition. This assay showed a correlation close to the gold standard plaque reduction neutralization test[39].

**IFN-γ release and quantification**. The SARS-CoV-2 IGRA stimulation tube set (ET 2606-3003, EUROIMMUN) was used according to the manufacturer's instructions to stimulate T-cells against S1 protein peptides. Briefly, 0.5 ml of heparinized whole blood was transferred into three different stimulation tubes: a blank tube for background levels of interferon-gamma (IFN-γ) from unstimulated T-cells, a test tube containing peptides derived from the S1 subunit of S protein, and a positive tube containing a mitogen used as a positive control of stimulated T-cells. Stimulation tubes were inverted six times (or more if necessary) until the surface was completely covered by blood and they were incubated at 37 °C for 21 h. Subsequently, tubes were centrifuged at $12,000 \times g$ for 10 min to stop the reaction. Plasma was collected, aliquoted, and stored at −80 °C until IFN-γ determination.

IFN-γ ELISA kit (ET 6841-9601, EUROIMMUN) was employed to quantitatively determine IFN-γ levels released after stimulating T-cells against S1 protein according to manufacturer's instructions. Briefly, samples were diluted at 1:5, 1:20, or 1:50 in sample buffer and incubated for 120 min. A standard curve and two controls were added undiluted and incubated, as mentioned before. IFN-γ was detected using a biotin-labeled anti-IFN-γ antibody followed by a peroxidase-labeled streptavidin (30 min incubations). TMB was used as a substrate to develop the reaction for 20 min. Then, 0.5 M $H_2SO_4$ was added to stop the reaction. OD was measured as mentioned above. Plates were washed five times between steps and washing buffer was left for 30 s between washes. All incubation steps were performed at RT without shaking. The threshold for assay positivity was set to 200 mIU/ml.

**Statistical analyses and modeling**. Statistical analyses were performed using R (version 4.1.0 for Windows, R Foundation for Statistical, Computing). Estimation of antibody levels was performed using GraphPad Prism version 9.2.0 (GraphPad Software, La Jolla, CA). IgM, IgG, and IgA levels were interpolated using non-linear regression four-parameter curve fitting. Results were given in AU/ml, with 200 AU/ml being the antibody concentration at the lowest dilution of the calibrator. Statistical differences between days between vaccine doses were assessed using the Mann–Whitney U test. Associations between IFN-γ and age, sex, previous SARS-CoV-2 infection, IgG levels, and IgA responses were assessed using multiple linear regression. IFN-γ correlation with IgG levels was analyzed using the Spearman rank correlation test. IgG levels were represented and analyzed using a zero-inflated Gaussian mixed model. The days from the first vaccination were represented with four natural cubic splines to allow the modeling of non-linear trends. The changes in antibody levels were modeled from the time of the first vaccine administration and up to 230 days. Assuming the decline rate of IgG maintains the same after reaching the maximum peak, we projected the levels of IgG antibodies up to 365 days after vaccination. Additionally, a linear mixed model was used to analyzed IgG levels from seven days after the second vaccine administration. IgM and IgA levels and the neutralization index were represented and analyzed using a generalized mixed model with a binomial distribution. IgM and IgA levels were transformed into a binary variable defined as positive or non-positive responses. Positive responses were defined as antibody values >200 AU/ml and >100 AU/ml of IgM and IgA, respectively. The neutralization index was transformed into a binary variable defined as neutralizing or non-neutralizing antibodies. Neutralizing antibodies were defined as >25% inhibitory capacity. Interactions were analyzed between days and previous infection (N protein positive), days and age groups (<40, 40–60, >60 years), and days and vaccine administrated (BNT162b2, ChAdOx1/

BNT162b2), and previous infection and age groups. Sex was included in the analysis as a covariate. Interactions of IgG levels over time with BMI (underweight (<18.5) normal (18.5–24.9), overweight (25–29.9) obese (>30)) as well as interactions between IgG levels and IgA responses over time and interval between the first and the second dose of the BNT162b2 vaccine (divided by tertiles into <29, 29–31, >31 days) were analyzed separately. IFN-γ released from T-cells (divided by tertiles into low, intermediate, and high levels) at approximately 6 months was included as an interaction for the IgG analyses using the same model used for IgG antibodies. For all analyses, IgG levels were log10 transformed and back-transformed when reported. *P*-values were calculated using Type II Wald χ-square tests, and a *p*-value <0.05 was considered significant. A more detailed description of the models can be found in the supplementary information.

**Reporting summary**. Further information on research design is available in the Nature Research Reporting Summary linked to this article.

## Data availability

The data used in this study are available in the Zenodo database [https://doi.org/10.5281/zenodo.6234161].

## Code availability

The code used in this study is available in the Zenodo database [https://doi.org/10.5281/zenodo.6234433].

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

## Acknowledgements

This work was financially supported by grants from the Carlsberg Foundation (CF20-476 0045, granted to P.G.), the Novo Nordisk Foundation (NFF205A0063505 and NNF20SA0064201, granted to P.G.), and the Svend Andersen Research Foundation (SARF2021, granted to P.G.). The Danish COVID-19 Biobank (part of Bio- and Genome Bank Denmark) is acknowledged for biological material and for data regarding handling and storage. Novo Nordisk A/S is acknowledged for their help producing recombinant proteins. The authors would like to thank Mads Engelhardt Knudsen, Sif Kaas Nielsen and Bettina Eide Holm from the Laboratory of Molecular Medicine at Rigshospitalet, Betina Poulsen from The Blood Bank, Department of Clinical Immunology, Rigshospitalet, Lisbeth Andreasen, Annie Mørk, Fie Andreasen, Ann Kristine Thorsteinsson, Tung Thanh Phan, and Ida Stenroos-Dam from the Department of Clinical Biochemistry at Rigshospitalet, for their excellent technical assistance in processing and analyzing the samples. We would also like to thank Alexandra Rosengaard Röthlin Eriksen from the Department of Emergency medicine, Herlev and Gentofte Hospital, for her logistics and sample collection assistance.

## Author contributions

K.K.I., H.B., S.D.N., and P.G. conceived and designed the study. L.P.-A., C.B.H., I.J., R.F.-S., and L.M.H. performed experiments; L.P.-A., J.J.A.A., J.R.M., S.B., and P.G. analyzed the data; S.R.H., L.D.H., M.M.P.H., D.L.M., K.F., R.B.H., H.B., S.D.N., and K.K.I. collected samples and clinical data. A.R. and R.B.-O. enabled recombinant protein production; E.S., M.A.H.L., and S.R.O. were in charge of biobanking. L.P.-A., J.J.A.A., and P.G. wrote the manuscript with subsequent inputs from the co-authors. All co-authors approved the final version of the manuscript.

## Competing interests

The authors declare no competing interests.
