## [Peer Review File · Nature Communications]

Modeling of waning immunity after SARS-CoV-2 vaccination and influencing factorsREVIEWER COMMENTS

Reviewer #1 (Remarks to the Author):

Limitations of this reviewer:

Before diving in, I like to try to describe what I think I know well & what I don't in hopes of creating more frank discussions that distinguish between informed reviews and naïve questions for the authors. Additionally, because I value direct feedback that identifies both parties in order to incentivize kindness and community, and in an effort to put the "peer" in "peer review", I provide my name & contact information should the authors want to ask for clarifications and/or iterate on any reviews I provide to save time in these time-intensive full submission + review iterations. I hope this is helpful!

My background is both math & biology, primarily focusing on forecasting in dynamical systems but also developing new statistical tools to study biological phenomena. I'm moderately conversant in immunology but the full understanding of the epidemiological implications of specific cytokines and all that molecular bio alphabet soup eludes me. With that said, I'll try to provide reviews that clarify when I feel I know the topic enough to know what I'm talking about, and when I'm curious but unsure.

I look forward to reviewing this paper!
-Alex Washburne

Summary of work:

The authors study the timeline of waning immunity by collecting, analyzing and modelling the IgG levels, IgM + IgA positivity, neutralization ability as time elapses since vaccination, and the associations between IgG and IFN-gamma.

The paper is clear, concise, and doesn't have any major methodological flaws. While I think some claims about their predictive models are overconfident (in the statistical sense, not the personal sense), I think those issues are easily addressed and the paper adds value to the field.

Major points:

In my opinion, this article should be published – the question of waning immunity is of central importance to our current understanding of Delta outbreaks and our forecasting & public health responses for future outbreaks. The more different ways the scientific community views this elephant of an issue like blind people with empirical/statistical hands, the faster we'll understand this crucial topic. I provide one major revision below but want to emphasize that the proposed discussion/revision doesn't really detract from an excellent paper – the figures are clear (mostly – see minor revisions), the methods are organized & good enough, and the benefit of getting these findings into print for others to think about exceeds the potential costs of slightly-imperfect out-of-sample predictions mentioned below (the only "costs" of such imperfect modelling would occur if people took the out-of-sample projections, in both mean & confidence intervals, too literally... some of that is avoided by the authors acknowledging the difficulty/ambiguity in connecting these immunological metrics to population-level epidemiological outcomes).

The main revision I recommend is to consider limitations in the use of GAMs as forecasting tools, especially GAMs that use piecewise cubic splines as their basis. The GAM's used are fantastic, versatile tools for estimating canonical parameters within our observed sample space, but their performance beyond our observed space (e.g. times beyond the time-series observed) can be quite poor in many common settings, and I believe this setting is one of them. For any timeseries that we believe converges to some horizontal asymptote (as I hypothesize all immunological assays do, absent subsequent vaccinations/infections), cubic splines that explode to +/- infinity

are guaranteed to perform poorly at some point in the future and we don't know when that point is. Looking closely at figure S4, for example, there's concavity in the latest timepoints such that I suspect if one fit a GAM up to 150 days and predicted the next 50-100 days, their predictions would underestimate IgG levels. Given that difficult-to-model curvature that we see exists at the latest timepoints, and given this curvature is one of the features that can cause GAMs to perform particularly poorly out-of-sample, I suspect a different approach, such as a dynamic linear model, can produce better-calibrated out-of-sample statements both by wider confidence intervals and explicit uncertainty around that end-of-series curvature reflecting our uncertainty in the out-of-sample curvature changes. DLMS have their own challenges and limitations (e.g. I'm not immediately familiar with a package that makes random effects easy in DLMS), and in my opinion GAMs are sufficiently familiar to biologists that they are often adequate (folk in epidemiology forecast with GAMs all the time, though we almost never use them for long time-horizon forecasts in finance for the issues mentioned above, and by "long" I mean any time horizon over which the most recent third derivative of our time series is significantly non-zero... hence my look at a model fit to 150 days projected to 200-250 days).

With all this said, it's perfectly fine to publish the paper with these GAMs but I wanted to bring this point up to the authors in case they wanted to reconsider their forecasts/projections far out-of-sample and/or tone down their use of "predictive model" as a keyword in this paper (it's a great inferential model, but almost surely biased predictive model). Figure 1 has data ending at ~200 days, a 75% quantile knot probably around 150- days, and projections shooting beyond 350 days – that's far enough out-of-sample that I'd be cautious about the accuracy of a GAM's estimate of the mean IgG levels and suspect the confidence intervals are too narrow.

While I've used a lot of words describing this issue, I want to use some more to reiterate that this limitation isn't that big of a deal and shouldn't detract from the very excellent study design, sample collection, and sound analyses. I still extracted very useful information out of this paper by seeing the raw data visualizations and (IMO more appropriately) using the GAM's to understand in-sample patterns instead of out-of-sample predictions. Mainly, I see this limitation as a way to either tone down some language about "prediction" and advertising this as a "predictive model", as most who do forecasting focus on rolling out-of-sample performance metrics that help us evaluate the stability of the signal, and not the usual goodness-of-fit for in-sample data as justification for projecting beyond our dataset. Note, this challenge is amplified by the final knot at the 3rd quartile, as the projections beyond our dataset are really just leveraging data in the final quartile (maybe a 50-day range?) to project 150 days later. Put differently, I'm much less certain about the out-of-sample forecasts than the authors' GAMs are, and consequently less certain about claims on the duration of immunity, time-until-seronegativity, etc. Again, while overhauling the GAMs or reframing the study to downplay predictions could be a major revision, it doesn't detract from what I believe is an excellent study.

The authors are free to leave all their models as-is (in general, I like seeing people doing things differently than I'd do them – that sort of methodological diversity keeps us all honest and adds value to science IMO). If so, the easiest solution that would reduce my own skeptical-reaction/major-revision here would be to just tone down claims about this being a "predictive" model outside our time window, keep the figures as they are for readers to qualitatively imagine their own out-of-sample curves, and point out these limitations of GAMs in the limitations section of the discussion. A harder solution would be to e.g. run a DLM or a parametric curve justified by other analyses of seropositivity dynamics, or another forecasting/prediction-designed model that learns how uncertain we ought to be in the wiggles of curvature over this time series for projections outside of our observed time window.

I really liked the examination of different trajectories for N-positive/N-negative patients as well as the partitioning of these trajectories by age & vaccine & IFN-gamma levels.

If these data were able to be made publicly available, that would also reduce the scrutiny of your own models by allowing others to examine predictive models of their own (e.g. some might be interested in particular age-x-IFN related hypotheses, or other second-order effects that might inform their own studies in this regard).

That's all! Great study folks – I'm looking forward to seeing this in print.

Minor points:

Most figures: the legends are too small & made it a bit difficult to see (I assume this is correct) that dotted lines correspond to N-positive HCWs. I could see the colors well, just not the solid vs. dashed delineation.

Figure 4: the transparency & size makes it difficult to distinguish between circles & triangles.

Reviewer #2 (Remarks to the Author):

The novelty of this work relies on modelling the longevity of antibody levels post-vaccination and assessing some of the factors that may affect waning in a prospective cohort. The most attractive aspect of the approach is the possibility to predict personalized antibody level dynamics to find eligible candidates for booster vaccination, although we can only do this now for seroreversion but not to for the protective threshold as this is presently unknown, as acknowledged.

The main strengths are:

- The length of the follow up, of almost up to 8 months after dose 1
- Addition of IgA analysis, although not totally well integrated
- Addition of T cell work, although more limited and not highly dimensional

The weaknesses are:

- Among the factors assessed, there is a limited amount of covariates that may impact waning. Beyond age, sex and weight, the study lacks examination of the impact of comorbidities including treatment use and smoking, among others. Also, among pre-exposed, no study of the impact of symptoms (type, duration...)
- Among the antigens used, only RBD is analyzed. In studies using multiple proteins it is seen that the dynamics of S, S1, S2 and RBD are not identical, therefore a more powerful evaluation would include all those fragments that are included in spike vaccines, not only the RBD that be detected at lower levels or in less individuals than anti-S responses
- Heterogeneity of the schedules compared with different vaccines and variable timepoints, that does not allow head to head comparison of the two vaccines after a full schedule. Also, the lack of relevant vaccines like Moderna, J&J or others
- Lack of assessment of impact of VoCs on antibody responses and kinetics

Reviews needed:

Abstract requires careful review of English and style, and also other parts of the manuscript.

Introduction

- This statement "Whether a delay in the vaccine interval may affect the strength of the immune response and how long time the immunity may last is unknown" has no references, please update with the studies that are already showing data about strength of response with different intervals and homologous vs heterologous vaccination
- The analysis of IgA is not well integrated nor justified. There are multiple studies that assess IgA in blood, after natural infection or vaccination; less studies look at mucosal IgA particularly post i.m. vaccination (but this study does not do that either, it analyzes serum IgA). Please revise and add references.

Methods

- In this sentence (stats methods) "Assuming the decline rate of IgG maintains the same, we

predicted the levels of IgG titers up to 365 days after vaccination", do you refer that this is the date that levels cross the positivity threshold? If so, specify

- No other variables were collected that may impact the responses? also, for infection, the exact occupation (e.g. proximity to patients, prevention measures, ...) may impact risk and levels

- When presenting the percentage of antibodies that are lost for each vaccine according to model predictions, please specify the timeframe

Results

- Prior exposure status: the number of participants with prior N positive serology and prior RT-PCR positivity are given separately, it is not clear the overlap and correlation, and how many were symptomatic or asymptomatic

- This finding is different to other reports, please discuss "This suggests that longer intervals between vaccines (>60 days between ChAdOx1/BNT162b2) appear to mount a less robust antibody response". Also see below a related point.

Discussion

- Probably one of the most important and novel aspects of the study, and potentially controversial, which is impact of vaccination interval in the kinetics, is not clear enough: "Interestingly, we observed a significant difference in the IgG waning in individuals fully vaccinated with BNT162b2 who had a longer interval between vaccines." The delay appears to have affected the second AstraZeneca vaccine in Denmark but sentence refers to the Pfizer vaccine, please clarify. Also, other reports indicate better (peak) responses with longer intervals but they had not assessed impact on longevity. So, are data suggesting that longer intervals may be better for early but not later response? if so, please make this clearer and emphasize if data are robust enough to conclude this

- The paragraph starting with "The results of this real-world longitudinal study ..." requires revision of the language to avoid repetitions, use of punctuation and more streamlining.

- Discussion is somewhat long and fragmented by topics, it will benefit by more integration and streamlining, and some parts could be shortened.

Minor reviews:

Abstract

- It would be informative to include the time interval (median, range) between doses for each scheme

- These sentences appear disconnected, need to be clarified to what they refer to: "IgA only became significant in naturally infected. Sex and age did also influence". If there is space constraints, you may delete "Antibody responses were higher among those naturally infected before vaccination completion" as this has been already reported and this paper's novelty is about the kinetics and not as much the peak response post vaccination.

Introduction

- In this sentence clarify that protection is against disease: "the vaccines elicit a rapid and highly protective immune response"; later it is referred to severe disease, so you may combine the concepts to avoid repetition (reorder)

- Third sentence rewrite to avoid repeating "second dose"

Methods

- Avoid repeating "combine" words in "Participants received either two doses of the BNT162b2 COVID-19 vaccine from Pfizer-BioNTech (BNT162b2) or the combination of one dose of the ChAdOx1-nCoV19 from Oxford/AstraZeneca combined with one dose of the BNT162b2 COVID-19 vaccines (ChAdOx1/BNT162b2)."

Results

- Figure 1: y axis should be antibody levels, no titers (also in results narrative)
- Figure 4: the long title is confusing, rewrite please
- Supp Fig 1: not clear the intervals of the max-min periods in dose 1 and 2 for Pfizer, they appear to overlap fully for the left interval?

Supplementary

Methods: "Pre-processing of the data" does not really describe this. Either add the detailed data pre-processing description (how the ELISA raw data was used to obtain AU before statistical modelling) or change the title.

Other general reviews:

- Revise format, some symbols have disappeared
- Add spike as another keyword

Reviewer #3 (Remarks to the Author):

In this study Danish healthcare workers were followed for approximately 6 months following their first dose of a SARS-CoV-2 vaccine. A number of immunoassays are implemented to measure IgG, IgA and IgM antibody responses, and a measure of T-cell mediated immunity. A pseudo-neutralization assay is also used to provide a measure of functional immunity. These data are then analyzed using generalized mixed regression models with cubic splines to study the duration of the immune response. The key claims of this article are based on estimates of the duration of immunity for up to one year. I don't believe the combination of model and data presented here are appropriate for making these inferences.

Major comments

- The time span of the data was approximately 6 months after first dose. The authors then provide extrapolation of immune markers beyond the range of the data, up to 12 months. While it is possible to provide model-based predictions beyond the range of the data, cubic splines are not fit for this purpose.
- The temporal analysis of immune markers is based on time since first dose. Following the first dose, the most important determinant of immune markers is the timing of the second dose. For example, the cubic spline model picks up a second peak around day 130 (line 140) which will be attributable to the timing of the second dose. For a model that considers time since first dose, the timing of the second dose would need to be carefully included as an additional covariate. In fact, a more appropriate and frequently used approach is to analyse data based on time since second dose, and to incorporate the timing between doses as a covariate. See for example:

Aldridge et al. Waning of SARS-CoV-2 antibodies targeting the Spike protein in individuals post second dose of ChAdOx1 and BNT162b2 COVID-19 vaccines and risk of breakthrough infections: analysis of the Virus Watch community cohort.

<https://www.medrxiv.org/content/10.1101/2021.11.05.21265968v1>

- In the study design, individuals administered ChAdOx1 as a first dose had a significantly longer duration to second dose (with BNT162b2) compared to individuals administered BNT162b2 as a first dose. This introduces important confounding between dose interval and vaccine interval.
- IgG levels and IFN-gamma levels are analyzed as continuous data. IgA and IgM data are analyzed as binary data. Given that all assays return a continuous output, this different choice of analysis methods is particular. Notably, the IgA and IgM are not presented in their continuous form.
- Data from pseudo-neutralization assays are presented as a measure of functional activity. At places this is mistakenly referred to as the neutralizing antibody (nAb). Although there may be a good correlation between neutralization and pseudo-neutralization, it is important to keep a clear distinction between these assays.
- In Figure 4, the y-axis label may be incorrect.

Reviewer #1 (Remarks to the Author):**Limitations of this reviewer:**

Before diving in, I like to try to describe what I think I know well & what I don't in hopes of creating more frank discussions that distinguish between informed reviews and naïve questions for the authors. Additionally, because I value direct feedback that identifies both parties in order to incentivize kindness and community, and in an effort to put the "peer" in "peer review", I provide my name & contact information should the authors want to ask for clarifications and/or iterate on any reviews I provide to save time in these time-intensive full submission + review iterations. I hope this is helpful!

My background is both math & biology, primarily focusing on forecasting in dynamical systems but also developing new statistical tools to study biological phenomena. I'm moderately conversant in immunology but the full understanding of the epidemiological implications of specific cytokines and all that molecular bio alphabet soup eludes me. With that said, I'll try to provide reviews that clarify when I feel I know the topic enough to know what I'm talking about, and when I'm curious but unsure.

I look forward to reviewing this paper!

-Alex Washburne

Summary of work:

The authors study the timeline of waning immunity by collecting, analyzing and modelling the IgG levels, IgM + IgA positivity, neutralization ability as time elapses since vaccination, and the associations between IgG and IFN-gamma.

The paper is clear, concise, and doesn't have any major methodological flaws. While I think some claims about their predictive models are overconfident (in the statistical sense, not the personal sense), I think those issues are easily addressed and the paper adds value to the field.

Major points:

Major Point 1: In my opinion, this article should be published – the question of waning immunity is of central importance to our current understanding of Delta outbreaks and our forecasting & public health responses for future outbreaks. The more different ways the scientific community views this elephant of an issue like blind people with empirical/statistical hands, the faster we'll understand this crucial topic. I provide one major revision below but want to emphasize that the proposed discussion/revision doesn't really detract from an excellent paper – the figures are clear (mostly – see minor revisions), the methods are

organized & good enough, and the benefit of getting these findings into print for others to think about exceeds the potential costs of slightly-imperfect out-of-sample predictions mentioned below (the only “costs” of such imperfect modelling would occur if people took the out-of-sample projections, in both mean & confidence intervals, too literally... some of that is avoided by the authors acknowledging the difficulty/ambiguity in connecting these immunological metrics to population-level epidemiological outcomes).

The main revision I recommend is to consider limitations in the use of GAMs as forecasting tools, especially GAMs that use piecewise cubic splines as their basis. The GAM’s used are fantastic, versatile tools for estimating canonical parameters within our observed sample space, but their performance beyond our observed space (e.g. times beyond the time-series observed) can be quite poor in many common settings, and I believe this setting is one of them. For any timeseries that we believe converges to some horizontal asymptote (as I hypothesize all immunological assays do, absent subsequent vaccinations/infections), cubic splines that explode to +/- infinity are guaranteed to perform poorly at some point in the future and we don’t know when that point is. Looking closely at figure S4, for example, there’s concavity in the latest timepoints such that I suspect if one fit a GAM up to 150 days and predicted the next 50-100 days, their predictions would underestimate IgG levels.

Given that difficult-to-model curvature that we see exists at the latest timepoints, and given this curvature is one of the features that can cause GAMs to perform particularly poorly out-of-sample, I suspect a different approach, such as a dynamic linear model, can produce better-calibrated out-of-sample statements both by wider confidence intervals and explicit uncertainty around that end-of-series curvature reflecting our uncertainty in the out-of-sample curvature changes. DLMS have their own challenges and limitations (e.g. I’m not immediately familiar with a package that makes random effects easy in DLMS), and in my opinion GAMs are sufficiently familiar to biologists that they are often adequate (folk in epidemiology forecast with GAMs all the time, though we almost never use them for long time-horizon forecasts in finance for the issues mentioned above, and by “long” I mean any time horizon over which the most recent third derivative of our time series is significantly non-zero... hence my look at a model fit to 150 days projected to 200-250 days).

With all this said, it’s perfectly fine to publish the paper with these GAMs but I wanted to bring this point up to the authors in case they wanted to reconsider their forecasts/projections far out-of-sample and/or tone down their use of “predictive model” as a keyword in this paper (it’s a great inferential model, but almost surely biased predictive model). Figure 1 has data ending at ~200 days, a 75% quantile knot probably around 150- days, and projections shooting beyond 350 days – that’s far enough out-of-sample that I’d be cautious about the accuracy of a GAM’s estimate of the mean IgG levels and suspect the confidence intervals are too narrow.

While I’ve used a lot of words describing this issue, I want to use some more to reiterate that this limitation isn’t that big of a deal and shouldn’t detract from the very excellent study design, sample collection, and sound analyses. I still extracted very useful information out of this paper by seeing the raw data visualizations and (IMO more appropriately) using the GAM’s to understand in-sample patterns instead of out-of-sample predictions. Mainly, I see this limitation as a way to either tone down some language about “prediction” and advertising this as a “predictive model”, as most who do forecasting focus on rolling out-of-sample performance metrics that help us evaluate the stability of the signal, and not the usual goodness-of-fit

for in-sample data as justification for projecting beyond our dataset. Note, this challenge is amplified by the final knot at the 3rd quartile, as the projections beyond our dataset are really just leveraging data in the final quartile (maybe a 50-day range?) to project 150 days later. Put differently, I'm much less certain about the out-of-sample forecasts than the authors' GAMs are, and consequently less certain about claims on the duration of immunity, time-until-seronegativity, etc. Again, while overhauling the GAMs or reframing the study to downplay predictions could be a major revision, it doesn't detract from what I believe is an excellent study.

The authors are free to leave all their models as-is (in general, I like seeing people doing things differently than I'd do them – that sort of methodological diversity keeps us all honest and adds value to science IMO). If so, the easiest solution that would reduce my own skeptical-reaction/major-revision here would be to just tone down claims about this being a “predictive” model outside our time window, keep the figures as they are for readers to qualitatively imagine their own out-of-sample curves, and point out these limitations of GAMs in the limitations section of the discussion. A harder solution would be to e.g. run a DLM or a parametric curve justified by other analyses of seropositivity dynamics, or another forecasting/prediction-designed model that learns how uncertain we ought to be in the wiggles of curvature over this time series for projections outside of our observed time window.

I really liked the examination of different trajectories for N-positive/N-negative patients as well as the partitioning of these trajectories by age & vaccine & IFN-gamma levels.

If these data were able to be made publicly available, that would also reduce the scrutiny of your own models by allowing others to examine predictive models of their own (e.g. some might be interested in particular age-x-IFN related hypotheses, or other second-order effects that might inform their own studies in this regard).

That's all! Great study folks – I'm looking forward to seeing this in print.

Major point 1 comment: We thank the reviewer for the thorough analysis of the results presented in this manuscript. We acknowledge that the prediction wording might have been too strong for the intention we had. The idea by showing the out-of-sample curves was not to make a strong statement about the prediction capabilities of the model. The aim was to illustrate how the levels of IgG would develop in the span of a year if the decay rate observed in the last period was maintained. We are aware that splines are not a powerful tool for data extrapolation, but in our case, we utilize natural cubic splines which extrapolate linearly beyond the range of the last knot (the limit of the data). We investigated whether it was possible to utilize forecasting models, such as DLM or ARIMA models. As pointed out, there are no implementations for random effects in such models, as we would have one trend per individual. DLMs are meant for covariates that are longitudinal and in our case, none of the covariates (N protein positivity, age, sex or first vaccine) change over time. However, methods for forecasting of grouped time series exists, such as an ARIMA model with a bottom-up approach. The difficulty that arises then is that the data is sampled irregularly across time, and this type of model requires equally-spaced observations across time. The only possibility would be to use more advanced machine learning methods that make us of recurrent neural networks and ordinary differential equations (ODEs) to model this type of irregularly sampled data. Due to the complexity of such models and difficulty

of interpretability, we decided that GLMM with natural cubic splines was the most suitable approach. We have toned down our prediction capabilities claims and leave it up to the reader to make their own interpretations about the out-of-sample trends. We have changed the plots to indicate where the out-of-sample trends start. Furthermore, we have included an additional linear model from the second vaccination dose (new Figure 2) to illustrate the differences between using linear vs non-linear models to fit IgG levels over time.

Minor points:

Minor point 1: Most figures: the legends are too small & made it a bit difficult to see (I assume this is correct) that dotted lines correspond to N-positive HCWs. I could see the colors well, just not the solid vs. dashed delineation.

Minor point 1 comment: We acknowledge the legends are difficult to see. We have increased the legend size to facilitate the reader to identify the labels and understand the figures.

Minor point 2: Figure 4: the transparency & size makes it difficult to distinguish between circles & triangles.

Minor point 2 comment: We have increased the size of the circles and triangles to facilitate the distinction between them (note: Figure 4 is now labeled as Figure 5).

Reviewer #2 (Remarks to the Author):

The novelty of this work relies on modelling the longevity of antibody levels post-vaccination and assessing some of the factors that may affect waning in a prospective cohort. The most attractive aspect of the approach is the possibility to predict personalized antibody level dynamics to find eligible candidates for booster vaccination, although we can only do this now for seroreversion but not to for the protective threshold as this is presently unknown, as acknowledged.

The main strengths are:

- The length of the follow up, of almost up to 8 months after dose 1
- Addition of IgA analysis, although not totally well integrated
- Addition of T cell work, although more limited and not highly dimensional

The weaknesses are:

Weakness 1: - Among the factors assessed, there is a limited amount of covariates that may impact waning. Beyond age, sex and weight, the study lacks examination of the impact of comorbidities including treatment use and smoking, among others. Also, among pre-exposed, no study of the impact of symptoms (type, duration...)

Weakness 1 comment: We acknowledge the limitation in the diversity of covariates that may impact waning. Unfortunately, we did not have access to information regarding other comorbidities nor symptomatology within infection-primed participants. We describe these limitations in the discussion (lines 382–385).

Weakness 2: - Among the antigens used, only RBD is analyzed. In studies using multiple proteins it is seen that the dynamics of S, S1, S2 and RBD are not identical, therefore a more powerful evaluation would include all those fragments that are included in spike vaccines, not only the RBD that be detected at lower levels or in less individuals than anti-S responses

Weakness 2 comment: We included only RBD in the study to analyze the antibody response towards the part of the protein S involved in the infection to the host cell upon binding to the ACE-2 receptor. Nevertheless, the reviewer rises an important point and we have included this statement as a limitation in the discussion section (lines 386–387).

Weakness 3: - Heterogeneity of the schedules compared with different vaccines and variable timepoints, that does not allow head to head comparison of the two vaccines after a full schedule. Also, the lack of relevant vaccines like Moderna, J&J or others

Weakness 3 comment: We acknowledge the lack of other relevant vaccines. At the moment of vaccination of the health care professionals and inclusion in the study, the vaccine stock bought by the Danish Authorities was primarily from Pfizer/BioNTech, followed by AstraZeneca/Oxford. Later in the vaccination strategy calendar, vaccines like Moderna and Johnson & Johnson were included in the vaccination strategy,

but it did not coincide with the inclusion time of our cohort. We have updated the limitations of our study regarding the lack of other vaccines in the discussion section (lines 388–390).

Weakness 4: - Lack of assessment of impact of VoCs on antibody responses and kinetics

Weakness 4 comment: The reviewer mentions an important point. The RBD used in this study was designed from the Wuhan strain, which is the one used for the basis of the currently approved vaccines. Nevertheless, and due to the great impact of the latest variant appeared, omicron, future studies would benefit the comparison of different RBDs from different VoCs. We have included this statement as a limitation of the study in the discussion section (lines 387–388).

Reviews needed:

Major point 1: Abstract requires careful review of English and style, and also other parts of the manuscript.

Major point 1 comment: We have revised the whole content of the manuscript and we have rewritten sentences and change wording to improve the language and style of the text.

Introduction

Major point 2: - This statement "Whether a delay in the vaccine interval may affect the strength of the immune response and how long time the immunity may last is unknown" has no references, please update with the studies that are already showing data about strength of response with different intervals and homologous vs heterologous vaccination

Major point 2 comment: We have updated the statement in lines 94–98, where we include references and outline the results from studies that have shown data about the delay in the vaccine interval in both homologous vaccination with BNT162b2 vaccine (reference 19 and 20) and heterologous vaccination with ChAdOx1/BNT162b2 vaccine (reference 21).

Major point 3: - The analysis of IgA is not well integrated nor justified. There are multiple studies that assess IgA in blood, after natural infection or vaccination; less studies look at mucosal IgA particularly post i.m. vaccination (but this study does not do that either, it analyzes serum IgA). Please revise and add references.

Major point 3 comment: The reviewer is correct about the lack of integration of the IgA analysis in the introduction. We have modified the paragraph in the introduction (lines 100–112), where we improve the rationale of the analysis of IgA antibodies in blood. Moreover, we have included a statement in the discussion to integrate better the analysis of IgA (lines 350–357).

Methods

Major point 4: - In this sentence (stats methods) "Assuming the decline rate of IgG maintains the same, we predicted the levels of IgG titers up to 365 days after vaccination", do you refer that this is the date that levels cross the positivity threshold? If so, specify

Major point 4 comment: In the above-mentioned sentence, we wanted to explain that we constructed the non-linear model assuming that the decrease in IgG levels follows same tendency after reaching the maximum peak of IgG antibodies. Then, we assume that the decline rate follows a unique linear trend and is constant for our projection in time up to 365 days after the first dose. We do not make reference when the antibody levels will cross the positivity threshold. We have updated this sentence (lines 492–493) to avoid any confusion to the reader.

Major point 5: - No other variables were collected that may impact the responses? also, for infection, the exact occupation (e.g. proximity to patients, prevention measures, ...) may impact risk and levels

Major point 5 comment: We understand the reviewer's concern but, unfortunately, for this study we collected information regarding age, sex, height, weight, number of doses administrated, name of the each vaccine dose administrated, date of administration of the first dose and date of administration of the second dose. Beyond this, we do not have information about the exact occupation of the health care professionals. We have included these limitations in the discussion (lines 382–385).

Major point 6: - When presenting the percentage of antibodies that are lost for each vaccine according to model predictions, please specify the timeframe

Major point 6 comment: We have included the timeframe in line 174, which is a period of 305 days from the peak of IgG levels (60 days) until the end of the model projection (365 days).

Results

Major point 7: - Prior exposure status: the number of participants with prior N positive serology and prior RT-PCR positivity are given separately, it is not clear the overlap and correlation, and how many were symptomatic or asymptomatic

Major point 7 comment: We acknowledge reviewer's observation that our statement about the infected-primed individuals in the results section can cause confusion. In our cohort, a total of 161 individuals were positive in the N-protein serology assay. Among these 161, 91 received a positive RT-PCR test before the first dose and 14 between the first and the second dose. A total of 56 participants did not have a RT-PCR test. We have modified the sentence in the results section (lines 130–132) to provide a clarification. As explained in the question regarding variables, we did not have access to the symptomatology of these individuals previously infected. We have included this limitation in the discussion section (lines 382–385).

Major point 8: - This finding is different to other reports, please discuss "This suggests that longer intervals between vaccines (>60 days between ChAdOx1/BNT162b2) appear to mount a less robust antibody response". Also see below a related point.

Major point 8 comment: We have clarified the statement in results in line 230 to explain better the finding. Based on our non-linear model we observed that individuals vaccinated with the ChAdOx1/BNT162b2 combination mounted a slightly lower peak both for IgG and IgA compared to the individuals vaccinated fully with BNT162b2. Moreover, the decline appeared to be more rapid in those who received the heterologous vaccine. Nevertheless, as explain below in the next point, there is a confounding effect between

vaccine type and interval between vaccines. So, we have toned down this claim in the discussion (lines 331–337).

Discussion

Major point 9: - Probably one of the most important and novel aspects of the study, and potentially controversial, which is impact of vaccination interval in the kinetics, is not clear enough: "Interestingly, we observed a significant difference in the IgG waning in individuals fully vaccinated with BNT162b2 who had a longer interval between vaccines." The delay appears to have affected the second AstraZeneca vaccine in Denmark but sentence refers to the Pfizer vaccine, please clarify. Also, other reports indicate better (peak) responses with longer intervals but they had not assessed impact on longevity. So, are data suggesting that longer intervals may be better for early but not later response? if so, please make this clearer and emphasize if data are robust enough to conclude this

Major point 9 comment: We acknowledge that the sentence might induce to confusion. Our intention was to evaluate if the difference in the IgG peak levels and the waning observed among the individuals fully vaccinated with BNT162b2 and those vaccinated with ChAdOx1/BNT162b2 was due to the vaccine interval or due to the different vaccination strategy. Although we have a confounding effect between vaccine interval and the vaccine administered, there are indications that a delay in the administration of the second dose also affects the peak levels and the stability of the IgG antibodies among individuals fully vaccinated with BNT162b2. This observation is stronger among infection primed individuals. To avoid confusions, we have rewritten the last part of the paragraph, stating the confounding situation, and clarifying the possible effect of the vaccine delay in the BNT162b2 vaccine (lines 331–337).

Major point 10: - The paragraph starting with "The results of this real-world longitudinal study ..." requires revision of the language to avoid repetitions, use of punctuation and more streamlining.

Major point 10 comment: We have revised the paragraph and modify it accordingly.

Major point 11: - Discussion is somewhat long and fragmented by topics, it will benefit by more integration and streamlining, and some parts could be shortened.

Major point 11 comment: We acknowledge that the discussion might be lengthy. Our intention was to provide a clear and detailed discussion about the different topics that our manuscript includes without confusing the reader. We have rewritten some sentences to provide a better overview of this section.

Minor reviews:

Abstract

Minor point 1: - It would be informative to include the time interval (median, range) between doses for each scheme

Minor point 1 comment: We thank the reviewer for this suggestion; we also consider informative to include these values in the abstract. However, we are limited by 150 words and we might delete information that is crucial to summarize the study.

Minor point 2: - These sentences appear disconnected, need to be clarified to what they refer to: "IgA only became significant in naturally infected. Sex and age did also influence". If there is space constraints, you may delete "Antibody responses were higher among those naturally infected before vaccination completion" as this has been already reported and this paper's novelty is about the kinetics and not as much the peak response post vaccination.

Minor point 2 comment: We have modified the sentences accordingly to provide a more coherent description (lines 59–61).

Introduction

Minor point 3: - In this sentence clarify that protection is against disease: "the vaccines elicit a rapid and highly protective immune response"; later it is referred to severe disease, so you may combine the concepts to avoid repetition (reorder)

Minor point 3 comment: We have reordered the sentence that refers to severe diseases and combined it to the sentence "the vaccines elicit a rapid and highly protective immune response" to avoid repetition (lines 70–71).

Minor point 4: - Third sentence rewrite to avoid repeating "second dose"

Minor point 4 comment: "Second dose" has been deleted to avoid repetition (line 71).

Methods

Minor point 5: - Avoid repeating "combine" words in "Participants received either two doses of the BNT162b2 COVID-19 vaccine from Pfizer-BioNTech (BNT162b2) or the combination of one dose of the ChAdOx1-nCoV19 from Oxford/AstraZeneca combined with one dose of the BNT162b2 COVID-19 vaccines (ChAdOx1/BNT162b2)."

Minor point 5 comment: We have substitute "combined" by "followed by" (line 425).

Results

Minor point 6: - Figure 1: y axis should be antibody levels, no titers (also in results narrative)

Minor point 6 comment: We have changed the term "titers" by "levels" at the y axis as well as in the narrative.

Minor point 7: - Figure 4: the long title is confusing, rewrite please

Minor point 7 comment: We have shortened the title to: Circulating IFN- γ levels modeled to circulating IgG levels against RBD in individuals vaccinated against SARS-CoV-2. Please, note that Figure 4 now is labelled as Figure 5.

Minor point 8: - Supp Fig 1: not clear the intervals of the max-min periods in dose 1 and 2 for Pfizer, they appear to overlap fully for the left interval?

Minor point 8 comment: We apology if Supp Fig 1 induces to confusion. The intervals of max-min only represent the 2nd doses for both homologous and heterologous vaccination. The upper one represents the 2nd dose after a 1st dose with BNT162b2 (both orange color) and the lower one represents the 2nd dose after a 1st dose with ChAdOx1/BNT162b2 (the first one in blue, the second in orange colors).

Supplementary

Minor point 9: Methods: "Pre-processing of the data" does not really describe this. Either add the detailed data pre-processing description (how the ELISA raw data was used to obtain AU before statistical modelling) or change the title.

Minor point 9 comment: We acknowledge that the title might confuse the reader with the data processing described in the statistical analysis in the methods section. As our intention in the supplement was to aware the reader about the data that was not included in the statistical analysis and extra information that should be disclosed, we have change the title to "Data exclusion and modification before statistical analysis".

Other general reviews:

Minor point 10: - Revise format, some symbols have disappeared

Minor point 10 comment: Regarding the symbols, we only included gamma as the Greek symbol γ . However, we wrote the complete name of the symbol in other situations as rho or chi. We did not include the Greek symbol in case during the submission some re-formatting could eliminate these symbols. We have included these symbols in the manuscript.

Minor point 11: - Add spike as another keyword

Minor point 11 comment: We have added "Spike protein" as another keyword.

Reviewer #3 (Remarks to the Author):

In this study Danish healthcare workers were followed for approximately 6 months following their first dose of a SARS-CoV-2 vaccine. A number of immunoassays are implemented to measure IgG, IgA and IgM antibody responses, and a measure of T-cell mediated immunity. A pseudo-neutralization assay is also used to provide a measure of functional immunity. These data are then analyzed using generalized mixed regression models with cubic splines to study the duration of the immune response. The key claims of this article are based on estimates of the duration of immunity for up to one year. I don't believe the combination of model and data presented here are appropriate for making these inferences.

Major comments

Major point 1: • The time span of the data was approximately 6 months after first dose. The authors then provide extrapolation of immune markers beyond the range of the data, up to 12 months. While it is possible to provide model-based predictions beyond the range of the data, cubic splines are not fit for this purpose.

Major point 1 comment: We acknowledge that the prediction wording might have been too strong for the intention we had. The idea by showing the out-of-sample curves was not to make a strong statement about the prediction capabilities of the model. The aim was to illustrate how the levels of IgG would develop in the span of a year if the decay rate observed in the last period was maintained. We are aware that splines are not a powerful tool for data extrapolation, but in our case, we utilize natural cubic splines which extrapolate linearly beyond the range of the last knot (the limit of the data). We explored other models for forecasting of time series, but currently there are no implementations that can deal with repeated measurements (random effects) and irregularly sampled data (no data for all time points). We have toned down our prediction capabilities claims and leave it up to the reader to make their own interpretations about the out-of-sample trends. We have changed the plots to indicate where the out-of-sample trends start.

Major point 2: • The temporal analysis of immune markers is based on time since first dose. Following the first dose, the most important determinant of immune markers is the timing of the second dose. For example, the cubic spline model picks up a second peak around day 130 (line 140) which will be attributable to the timing of the second dose. For a model that considers time since first dose, the timing of the second dose would need to be carefully included as an additional covariate. In fact, a more appropriate and frequently used approach is to analyse data based on time since second dose, and to incorporate the timing between doses as a covariate. See for example:

Aldridge et al. Waning of SARS-CoV-2 antibodies targeting the Spike protein in individuals post second dose of ChAdOx1 and BNT162b2 COVID-19 vaccines and risk of breakthrough infections: analysis of the Virus Watch community cohort.

<https://www.medrxiv.org/content/10.1101/2021.11.05.21265968v1>

Major point 2 comment: We have further analyzed the interval length between doses for individuals with both BNT162b2 doses. For this, we have included in our GLMM with natural cubic splines, the days between doses as a covariate (in three equally-sized group) (Supplementary Figure 4). Furthermore, we have included an additional linear model from the second vaccination dose (new Figure 2) to illustrate the differences between using linear vs non-linear models to fit IgG levels over time.

Major point 3: • In the study design, individuals administered ChAdOx1 as a first dose had a significantly longer duration to second dose (with BNT162b2) compared to individuals administered BNT162b2 as a first dose. This introduces important confounding between dose interval and vaccine interval.

Major point 3 comment: There is indeed a confounding effect between dose interval and administered vaccine. This happens because there is only one individual with 20 days between the ChAdOx1 dose and the BNT162b2 dose. The other individuals had a median of 81 days between doses and no less than 60 days between doses. On the other hand, there are only 20 individuals with more than 40 days between both BNT162b2 doses. This means that the differences that we observe between individuals administered ChAdOx1 as a first dose and individuals administered BNT162b2 as a first dose cannot be attributed to either the vaccine type or the length of the interval between doses. These differences would have been possible to study if there were individuals with ChAdOx1 and BNT162b2 with similar dose interval as individuals with only BNT162b2, but unfortunately this was not the case. To make clearer this confounding, the legends have been changed to include the median days between doses for the two groups of individuals. Furthermore, we have included a more complete analysis of the interval length between doses for individuals with both BNT162b2 doses (Supplementary Figure 4 and Supplementary Figure 5).

Major point 4: • IgG levels and IFN-gamma levels are analyzed as continuous data. IgA and IgM data are analyzed as binary data. Given that all assays return a continuous output, this different choice of analysis methods is particular. Notably, the IgA and IgM are not presented in their continuous form.

Major point 4 comment: We analyzed the IgG levels and IFN-gamma levels as continuous data as both are normally distributed (except IgG at baseline hence the zero-inflated model). The IgA and IgM levels were analyzed as binary (positive/negative response), due to the great number of samples below the positivity threshold, which did not represent a normal distribution. For the IgG data, samples below the positivity threshold were mostly concentrated at baseline. However, for the IgA and IgM data, samples below the positivity threshold appeared evenly across all the time points. This meant that IgG data was normally distributed beyond the baseline, but IgA and IgM data was not normally distributed at any given time point. The IgA and IgM distributions showed a clear binary response, we either detect or we do not detect the antibody.

This is the reason why we do not represent IgA and IgM in their continuous form. We initially treated IgA and IgM levels as continuous data, but the quality control showed a poor model fit, with very skewed residuals. Therefore, we implemented the binary analysis as it was the best model to fit and interpret these data.

Major point 5: • Data from pseudo-neutralization assays are presented as a measure of functional activity. At places this is mistakenly referred to as the neutralizing antibody (nAb). Although there may be a good correlation between neutralization and pseudo-neutralization, it is important to keep a clear distinction between these assays.

Major point 5 comment: We understand the reviewer concern about our pseudo-neutralization assay. We based all our statements on the antibody levels as the dynamic of the assay is less limited as it is observed for the pseudo-neutralization assay. Therefore, the data from this assay provides extra information but we are cautious about it and we acknowledge its limitation in the results section (lines 239-240) and in the discussion (lines 318–320).

Major point 6: • In Figure 4, the y-axis label may be incorrect.

Major point 6 comment: We acknowledge the confusion that Figure 4 (now labeled as Figure 5) can create. We express in the y-axis the IgG levels. The IFN-gamma levels are included as a covariate of three categories according to low, intermediate, and high levels. We modelled then the IFN-gamma levels to the IgG levels.

REVIEWERS' COMMENTS

Reviewer #2 (Remarks to the Author):

The authors have made a good job at addressing the concerns.

There are couple of things that require revision:

- In the expansion of the IgA literature in the Introduction, this sentence is not quite accurate "whereas the elicited IgA responses appear to be not efficient and rather disappear rapidly 25." Please review well prior work as, contrary to earlier expectations, IgA levels can also last. The limitation here is that authors only included RBD that may last shorter but IgA to full spike IgA can last longer and indeed does not disappear rapidly.

Related to this, be careful with speculations in discussion, line 354. If mucosal IgA is not measured (only serum) and considering that i.m. vaccination is not supposed to induce this local response (only oral/inhaled/nasal), there is no need to interpret the lack of anti-infection vaccine effect on the basis of mucosal IgA. I would restrict the paper to the kinetics as this is tangential in the study.

- Minor: line 184 correct "personalized". In general, the new track changes text requires another revision for English/style

Reviewer #3 (Remarks to the Author):

The authors have made some notable and worthwhile revisions to the manuscript. However, they side-stepped two major points from my original review that I consider to be amongst the most important.

Major point 1: The time span of the data was approximately 6 months after first dose. The authors then provide extrapolation of immune markers beyond the range of the data, up to 12 months. While it is possible to provide model-based predictions beyond the range of the data, cubic splines are not fit for this purpose.

Major point 1 comment: We acknowledge that the prediction wording might have been too strong for the intention we had. The idea by showing the out-of-sample curves was not to make a strong statement about the prediction capabilities of the model. The aim was to illustrate how the levels of IgG would develop in the span of a year if the decay rate observed in the last period was maintained. We are aware that splines are not a powerful tool for data extrapolation, but in our case, we utilize natural cubic splines which extrapolate linearly beyond the range of the last knot (the limit of the data). We explored other models for forecasting of time series, but currently there are no implementations that can deal with repeated measurements (random effects) and irregularly sampled data (no data for all time points). We have toned down our prediction capabilities claims and leave it up to the reader to make their own interpretations about the out-of-sample trends. We have changed the plots to indicate where the out-of-sample trends start.

Major point 1 response: It is not appropriate to use cubic splines to extrapolate beyond the temporal range of the data. Reviewer #1 made essentially the same point. Yet, the authors continue to make long-term projections. For example, from the abstract:

"A one-year IgG projection revealed an initial two-phase response"

A vertical dashed line has been added at ~ 230 days to Figure 1 to indicate where the out-of-sample trend starts. Note that the same line is used for all four panels, even though some data sets have a much smaller temporal range.

These projections are not valid and should not be made at all.

As an additional note, there are well established statistical methods for handling random effects with repeated measurements and irregularly sampled data. This is a specialty of the field of pharmacokinetics/pharmacodynamics (Pk/Pd) modelling. See for example the following book on the subject:

Marc Lavielle. Mixed Effects Models for the Population Approach: Models, Tasks, Methods and Tools

Major point 4: • IgG levels and IFN-gamma levels are analyzed as continuous data. IgA and IgM data are analyzed as binary data. Given that all assays return a continuous output, this different choice of analysis methods is particular. Notably, the IgA and IgM are not presented in their continuous form.

Major point 4 comment: We analyzed the IgG levels and IFN-gamma levels as continuous data as both are normally distributed (except IgG at baseline hence the zero-inflated model). The IgA and IgM levels were analyzed as binary (positive/negative response), due to the great number of samples below the positivity threshold, which did not represent a normal distribution. For the IgG data, samples below the positivity threshold were mostly concentrated at baseline. However, for the IgA and IgM data, samples below the positivity threshold appeared evenly across all the time points. This meant that IgG data was normally distributed beyond the baseline, but IgA and IgM data was not normally distributed at any given time point. The IgA and IgM distributions showed a clear binary response, we either detect or we do not detect the antibody. This is the reason why we do not represent IgA and IgM in their continuous form. We initially treated IgA and IgM levels as continuous data, but the quality control showed a poor model fit, with very skewed residuals. Therefore, we implemented the binary analysis as it was the best model to fit and interpret these data.

Major point 4 response: There can be good reasons to analyze data in a binary format. Without being able to see the IgA or IgM data, it is impossible for a reviewer or reader to know if this was a reasonable choice. The data is not plotted or publicly available. The following rather weak data access statement is made:

"Data are available upon reasonable request to the corresponding author."

The authors make comments regarding positivity thresholds, but the selection of a positivity threshold can be rather arbitrary. For example, I could choose a higher positivity threshold for the clearly continuous IgG data, and then claim the IgG also needs to be analyzed as binary data.

For the authors' information, continuous data on anti-IgA and anti-IgM antibody kinetics has been published, and made publicly available, since very early on in the COVID-19 pandemic. See for example the following:

- Persistence and decay of human antibody responses to the receptor binding domain of SARS-CoV-2 spike protein in COVID-19 patients (PMID: 33033172)
- Immunological memory to SARS-CoV-2 assessed for up to 8 months after infection (PMID: 33408181)
- Persistence of serum and saliva antibody responses to SARS-CoV-2 spike antigens in COVID-19 patients (PMID: 33033173)

Reviewer #2 (Remarks to the Author):

The authors have made a good job at addressing the concerns.

Point 1:

There are couple of things that require revision:

- In the expansion of the IgA literature in the Introduction, this sentence is not quite accurate "whereas the elicited IgA responses appear to be not efficient and rather disappear rapidly 25."

Please review well prior work as, contrary to earlier expectations, IgA levels can also last. The limitation here is that authors only included RBD that may last shorter but IgA to full spike IgA can last longer and indeed does not disappear rapidly.

Related to this, be careful with speculations in discussion, line 354. If mucosal IgA is not measured (only serum) and considering that i.m. vaccination is not supposed to induce this local response (only oral/inhaled/nasal), there is no need to interpret the lack of anti-infection vaccine effect on the basis of mucosal IgA. I would restrict the paper to the kinetics as this is tangential in the study.

Point 1 comment: We understand the reviewer's concern regarding the duration of circulating IgA antibodies. We have reviewed the latest literature and we acknowledge we might have anticipated the decrease of detectable circulating IgA antibodies. We have included two new references (25 and 26, line 106) showing that IgA antibodies against S1 protein are still detectable in some individuals up to 6 months; nevertheless, a significant decrease is observed. This tendency reminds similar to our results using RBD as an antigen, although we cannot claim anything else than a significant decrease over time.

We acknowledge that our speculations regarding vaccines and mucosal IgA might be out of the scope of the study. We have decided to eliminate the sentence "Thus the lack of a robust IgA response could be the explanation to why the current vaccines used only confer modest protection against transmission due to low mucosal responses. At the same time, they are highly effective in protecting against severe disease because of a robust systemic response" from the discussion (line 344).

Point 2:

- Minor: line 184 correct "personalized". In general, the new track changes text requires another revision for English/style

Point 2 comment: We appreciate the linguistic corrections indicated by the reviewer. We have corrected the word and checked the rest of the text for typos and style.

Reviewer #3 (Remarks to the Author):

The authors have made some notable and worthwhile revisions to the manuscript. However, they side-stepped two major points from my original review that I consider to be amongst the most important.

Major point 1 from first revision:

Major point 1 (from the reviewer, first revision): The time span of the data was approximately 6 months after first dose. The authors then provide extrapolation of immune markers beyond the range of the data, up to 12 months. While it is possible to provide model-based predictions beyond the range of the data, cubic splines are not fit for this purpose.

Major point 1 comment (from the authors, first revision): We acknowledge that the prediction wording might have been too strong for the intention we had. The idea by showing the out-of-sample curves was not to make a strong statement about the prediction capabilities of the model. The aim was to illustrate how the levels of IgG would develop in the span of a year if the decay rate observed in the last period was maintained. We are aware that splines are not a powerful tool for data extrapolation, but in our case, we utilize natural cubic splines which extrapolate linearly beyond the range of the last knot (the limit of the data). We explored other models for forecasting of time series, but currently there are no implementations that can deal with repeated measurements (random effects) and irregularly sampled data (no data for all time points). We have toned down our prediction capabilities claims and leave it up to the reader to make their own interpretations about the out-of-sample trends. We have changed the plots to indicate where the out-of-sample trends start.

Major point 1 response (from the reviewer, final revision): It is not appropriate to use cubic splines to extrapolate beyond the temporal range of the data. Reviewer #1 made essentially the same point. Yet, the authors continue to make long-term projections. For example, from the abstract:

“A one-year IgG projection revealed an initial two-phase response”

A vertical dashed line has been added at ~ 230 days to Figure 1 to indicate where the out-of-sample trend starts. Note that the same line is used for all four panels, even though some data sets have a much smaller temporal range.

These projections are not valid and should not be made at all.

As an additional note, there are well established statistical methods for handling random effects with repeated measurements and irregularly sampled data. This is a specialty of the field of

pharmacokinetics/pharmacodynamics (Pk/Pd) modelling. See for example the following book on the subject: Marc Lavielle. Mixed Effects Models for the Population Approach: Models, Tasks, Methods and Tools

Major point 1 comment (from the authors, final revision):

We understand reviewer's concern about certain limitations of the non-linear model included in this study. We would like to highlight that we do not use cubic splines, but natural cubic splines. This method extrapolates linearly beyond the range of the last knot (the limit of the data). Assuming the decreasing rate unaltered after the limit of the data, the projection would be reflected linearly and would not deviate as could occur when using polynomial cubic splines. The use of natural cubic splines allows to include the antibody levels from baseline, providing a novel method to study the complete response of vaccination, including several factors such as previous infection, age, etc. and its projection over time. We consider it is an advantage compared to more conservative models as the linear model that only takes into consideration antibody levels from the second dose and other factors for projections. We acknowledge it was beneficial to include the linear model in the study to compare the non-linear model with a more traditional one.

Upon further reinspecting Reviewer 1 comments regarding possible shapes of the antibody waning trajectory we contemplated that he might be referring to the prediction intervals, which reflect the uncertainty of future values and are usually wider than the confidence intervals. For further clarifications, we have included a version of Figure 1 in the Supplementary Information where the prediction intervals are depicted (Supplementary Figure 3).

Since the projection values appear to reflect a reasonable assumption about antibody waning over the projected time period we decided to keep the non-linear model in the manuscript but we have tone-down the sentence in the abstract (line 62) and included in the discussion section an additional note regarding the possibility of antibody waning taking different shapes (line 270), including the new Supplementary Figure 3 showing the uncertainty of future values.

We decided to use the same line across panels (vertical dashed line at day 230) because the same model is used to generate the curves across all panels. Nevertheless, to avoid any confusion we have modified the vertical lines to reflect the temporal range for each panel.

Major point 4 from first revision:

Major point 4 (from the reviewer, first revision): IgG levels and IFN-gamma levels are analyzed as continuous data. IgA and IgM data are analyzed as binary data. Given that all assays return a continuous output, this different choice of analysis methods is particular. Notably, the IgA and IgM are not presented in their continuous form.

Major point 4 comment (from the authors, first revision): We analyzed the IgG levels and IFN-gamma levels as continuous data as both are normally distributed (except IgG at baseline hence the zero-inflated model). The IgA and IgM levels were analyzed as binary (positive/negative response), due to the great number of samples below the positivity threshold, which did not represent a normal distribution. For the IgG data, samples below the positivity threshold were mostly concentrated at baseline. However, for the IgA and IgM data, samples below the positivity threshold appeared evenly across all the time points. This meant that IgG data was normally distributed beyond the baseline, but IgA and IgM data was not normally distributed at

any given time point. The IgA and IgM distributions showed a clear binary response, we either detect or we do not detect the antibody.

This is the reason why we do not represent IgA and IgM in their continuous form. We initially treated IgA and IgM levels as continuous data, but the quality control showed a poor model fit, with very skewed residuals. Therefore, we implemented the binary analysis as it was the best model to fit and interpret these data.

Major point 4 response (from the reviewer, final revision): There can be good reasons to analyze data in a binary format. Without being able to see the IgA or IgM data, it is impossible for a reviewer or reader to know if this was a reasonable choice. The data is not plotted or publicly available. The following rather weak data access statement is made:

“Data are available upon reasonable request to the corresponding author.”

The authors make comments regarding positivity thresholds, but the selection of a positivity threshold can be rather arbitrary. For example, I could choose a higher positivity threshold for the clearly continuous IgG data, and then claim the IgG also needs to be analyzed as binary data.

For the authors' information, continuous data on anti-IgA and anti-IgM antibody kinetics has been published, and made publicly available, since very early on in the COVID-19 pandemic. See for example the following:

- Persistence and decay of human antibody responses to the receptor binding domain of SARS-CoV-2 spike protein in COVID-19 patients (PMID: 33033172)
- Immunological memory to SARS-CoV-2 assessed for up to 8 months after infection (PMID: 33408181)
- Persistence of serum and saliva antibody responses to SARS-CoV-2 spike antigens in COVID-19 patients (PMID: 33033173)

Major point 4 comment (from the authors, final revision): We understand reviewer's concern regarding the absence of a plot showing the data of IgM and IgA over time on the manuscript. Therefore, we have included an additional plot showing the values for IgG, IgA and IgM over time (Supplementary Figure 4). As can be observed, IgG goes quickly from non-detectable levels to detectable levels (relative frequency histogram below the observations), being above the threshold for the rest of the period. We can also see that there are almost no observations with zero value for IgG. IgA and IgM show a vastly different dynamic over time compared to IgG. We can see that most of the observations are below the positivity threshold and there is a much higher proportion of observations with zero value. This was the main issue when trying to model these observations as continuous. The high proportion of zeros breaks the assumptions of a normally distributed response variable thus producing very skewed non-normally distributed residuals. Therefore, we decided to model the IgA and IgM levels as a binary variable. When modelling this data under a binomial distribution the residuals were normally distributed meaning that the binary variable fitted much better the assumptions of the model. One could argue against our approach and advocate for modelling IgA and IgM as

continuous data by discarding the observations with zero value. We believe that this method would be erroneous as this would bias the analysis towards the individuals with higher levels of IgM and IgA. This means that we would only study these dynamics on individuals with high antibodies levels, which is not representative of the entire cohort (as most of the samples are zero or below the positivity threshold). Additionally, removing zero-valued observations would leave 180 and 254 individuals with three or more observations for the IgA and IgM analysis, respectively.

Regarding the selection of the positivity threshold, we based this value on the assay validation, which we made publicly available previously (SARS-CoV-2 Antibody Responses Are Correlated to Disease Severity in COVID-19 Convalescent Individuals, PMID: 33208457). The threshold value has not been modified, excluding the possibility of increasing or decreasing the positivity rate of the samples analyzed, remaining the same independently of the cohort analyzed.